# Addressing cultural and knowledge barriers to enable preclinical sex inclusive research

**Brianna N Gaskill¹†, Benjamin Phillips²†, Jonathan Ho³, Holly Rafferty⁴, Oladele Olajide Onada⁵, Andrew Rooney⁶, Amrita Ahluwalia³†, Natasha A Karp²*†**

[1]Novartis Biomedical Research, Frederick, United States; [2]Data Sciences and Quantitative Biology, Discovery Sciences, R&D, AstraZeneca, Cambridge, United Kingdom; [3]Faculty of Medicine & Dentistry, Queen Mary University of London, London, United Kingdom; [4]University of Glasgow, Glasgow, United Kingdom; [5]Obafemi Awolowo University and Obafemi Awolowo University Teaching Hospitals Complex, Ile-Ife, Nigeria; [6]Institute of Pharmacy and Biomedical Science, University of Strathclyde, Glasgow, United Kingdom

**\*For correspondence:**
Natasha.Karp@astrazeneca.com

†These authors contributed equally to this work

## eLife Assessment

The authors quantified intentions and knowledge gaps in scientists' use of sex as a biological variable in their work, and used a workshop intervention to show that while willingness was high, pressure points centered on statistical knowledge and perceived additional monetary costs to research. These **important** findings demonstrate the difficulty in changing understanding: while interventions can improve knowledge and decrease perceived barriers, the impact was small. The evidence for the findings is **solid**.

**Abstract** For over 30 years, research has highlighted a sex bias in early research, risking the validity of biological knowledge. The first step towards change is effectively challenging misconceptions, allowing researchers to perceive sex inclusive research as doable. Utilising the theory of planned behaviour, we quantified researchers' intention as a proxy measure for conducting sex inclusive research and explored attitude (value of the behaviour), subjective norm (perceived social pressure), and behavioural control (ability to conduct the behaviour). Additionally, we quantified the knowledge gap, prevalence of misconceptions, and assessed perceived benefits and barriers. We tested a workshop intervention that directly challenges the cultural embedded barriers. The data shows researchers' intentions were high, but they had weak statistical knowledge and misunderstandings leading to a perception that inclusive research is prohibitive due to cost and animal use. We demonstrate that participation in the training intervention improved knowledge, altered the perceived barriers, and cultural expectations.

## Introduction

Research culture encompasses the values, ideals, norms, and thus the resultant behaviours of a community. This resulting culture, in turn, has a substantial impact on how research is conducted and communicated. There is a long-established embedded practice of studying only a single (typically male) sex in preclinical research and then generalising the results to a wider population (*Beery and Zucker, 2011*; *Berkley, 1992*; *Woitowich et al., 2020*).

Since the 1990s, it has been recognised that this approach is highly limited (*Nunamaker and Turner, 2023*), with an emerging body of published evidence highlighting that sex may profoundly influence biological response (*Karp et al., 2017*; *U.S. Government Accountability Office, 2001*). Consequently, there has been growing awareness that unless sex is factored into experimental design, its influence remains unknown. Importantly, it is now appreciated that single-sex studies result in a knowledge bias (*Mogil, 2020*; *Clayton, 2016*). The need to drive cultural change has been highlighted, with the goal of enabling equitable research at the exploratory stage (*Rich-Edwards and Maney, 2023*) which delivers a translational body of knowledge relevant to the broader population (*Shansky and Murphy, 2021*).

The Sex and Gender Equity in Research (SAGER) guidelines (*Heidari et al., 2016*) provide guidance that 'Where the subjects of research comprise organisms capable of differentiation by sex, the research should be designed and conducted in a way that can reveal sex-related differences in the results, even if these were not initially expected'. This is a default position of inclusion where the sex can be determined and an analysis assessing for sex-related variability in response. Historically, sex has, of course, been included in studies where sex is a primary variable of interest (confirmatory sex inclusion) (*Karp, 2025*). The additional attention is focusing on sex inclusion in research studies where sex is not the primary variable of interest (exploratory sex inclusion). The goal of inclusion in these studies is to assess the generalisability of the treatment effect (*Karp, 2025*). As the objectives are very different, the underpinning design considerations are distinct. In exploratory sex inclusion, the N is shared between the sexes (*Phillips et al., 2023*). With this research strategy, if the effect is very different between the sexes, the statistical power passes from the main effect to the interaction term indicating that sex is significant at explaining variation in the treatment. This finding can then be used to generate sex-specific hypothesis and depending on the effect size and biology, a decision as to whether this should be explored further (*Karp, 2025*). Of course, when a research proposal has a hypothesis that is specific to a sex (e.g. it is related to oestrogen levels), it would be appropriate to only study the relevant sex. The recently published Sex Inclusive Research Framework (*Karp et al., 2025*) gives support to evaluate research proposals in terms of sex inclusion and allows an exemption for such a scenario classifying such proposals 'Single sex study justified'.

Despite numerous funding bodies establishing initiatives to encourage researchers to routinely integrate males and females into basic, preclinical, and clinical research, little progress has been observed (*Lee, 2018*; *Zucker et al., 2022*). As a response to the status quo, numerous funding bodies have recently issued inclusion mandates requiring justification for single-sex studies and analysis that considers sex-related variability in the response (*Rich-Edwards and Maney, 2023*; *Lee, 2018*; *Medical Research Council, 2022*). Positively, the proportion of published papers including females and males has improved (*Rechlin et al., 2022*; *Woitowich et al., 2020*). A comprehensive study comparing inclusion between 2009 and 2019 across nine biological disciplines found that six out of the nine disciplines saw improved representation, and overall inclusion improved from 26 to 48% of published studies (*Woitowich et al., 2020*). Whilst these studies demonstrate progress in inclusion, the proportion still represents a minority of studies, and notably, this inclusion was not associated with an increase in the proportion of studies that included data analysed by sex. Additionally, studies identified methodological issues when males and females were included. Examples include: a failure to report the proportion of the sexes in the cohort (*Stanford et al., 2023*) and inappropriate data analysis such as disaggregation, pooling, or comparison of *p* values to assess for treatment by sex interaction (*Rechlin et al., 2022*; *Garcia-Sifuentes and Maney, 2021*). There is a need to not only improve inclusion but also improve the analysis of the subsequent data to ensure the integrity of the conclusions made.

Notionally, scientists support efforts to improve sex representation in research (*Woitowich et al., 2020*; *Medical Research Council Working Group, 2022b*; *Woitowich and Woodruff, 2019*). However, sociological research suggests that many researchers believe inclusion is not 'doable' (*Gompers, 2018*; *Waltz et al., 2021*). The perceived barriers cited encompass ethical, financial, statistical, and practical constraints (*Medical Research Council Working Group, 2022b*; *Phillips et al., 2023*; *Karp and Reavey, 2019*). An analysis of 30 single-sex justifications identified six themes: concern around a known sex difference or sex effect (30%), fear of increased experimental variability (27%), experimental conditions limiting use of female and male animals (13%), limited sample size (13%), inability to sex subjects (10%), and issues with animal husbandry (7%) (*Woitowich et al., 2020*). Many of the barriers have been highlighted as culturally ingrained misconceptions (*Karp and Reavey, 2019*). For

example, a commonly proffered argument for exclusion is that female animals are inherently more variable, and thus inclusion will lead to the need for a larger sample size (N). This is despite a wealth of studies that reject this thesis of inherent increased variability across a range of biologically relevant endpoints (*Becker et al., 2016*; *Prendergast et al., 2014*; *Beery, 2018*). The Sex Inclusive Research Framework (*Karp et al., 2025*) provides support for nuanced thinking on this topic. First, it challenges the generic argument but allows for an individual proposal justification with project-relevant data, along with a cost-benefit analysis to support the use of single-sex studies where necessary. Another common misconception includes the perception that inclusion will require a doubling of the sample size as standard (*Waltz et al., 2021*; *Fields, 2014*; *Medical Research Council Working Group, 2022a*). Furthermore, many researchers have also questioned the value of studying multiple sexes in preclinical research, suggesting that single sex in vivo studies can be generalisable to the wider population (*Waltz et al., 2021*).

The scale of the misconceptions and statistical misunderstandings that result in barriers to inclusion have resulted in a call for training (*Rich-Edwards and Maney, 2023*; *Karp and Reavey, 2019*). We developed a workshop directed at equipping participants with knowledge of the current best practice for sex inclusion. The workshop included material designed to explicitly address many of the known misconceptions and barriers to sex-inclusive research. To evaluate the efficacy of this intervention, we ran two independent experiments using a survey to collect quantitative and qualitative data regarding self-efficacy (confidence to conduct inclusive research), knowledge, and views of scientists on the issues regarding incorporating males and females into preclinical in vivo research. The first experiment was conducted at an international conference with a parallel group design testing the general population attending the conference, those who expressed interest in the topic before attending a related seminar and those who attended the intervention. The second experiment was conducted at a UK Russell Group university with a pre- and post-intervention design. The survey was based on the theory of planned behaviour, as an established strategy for evaluating and understanding human behaviour (*Ajzen, 1991*). The theory of planned behaviour is a psychological theory that quantifies behavioural intention as a proximal determinant of behaviour through three core beliefs (attitude, subjective norms, and perceived behavioural control: See Appendix 9 for a glossary of these terms). We then assessed the impact of the intervention, with three key objectives: (1) to capture and quantify the scale of misconceptions around sex-inclusive studies within the research community, (2) to assess whether the intervention led to an improvement in knowledge, (3) to explore statistically the impact of the intervention on the intention to conduct sex-inclusive research, alongside identification of other predictors of intention. Taken together, the studies demonstrate that the intervention is effective and gives insight into how the research community can expedite change in research practice.

## Results

### Demographic results

#### Study 1

Across the three test groups, the demographics and distribution for the potential predictor variables were found to be balanced (*Supplementary file 1*). Furthermore, no significant correlation ($p>0.05$) was found between the continuous variables as assessed by a Pearson correlation analysis (*Appendix 1—table 1*). Missingness was observed in two demographic variables: two participants (Baseline: 1; Interested: 1) did not provide information on the years worked in animal research (missingness = 1.9%) and seven responders (Baseline: 4; Interested: 3) did not provide their age (missingness = 6.7%). The two who did not report years worked also did not report their age.

The majority of the participants were female (61%), had an average age of 39 years, had a PhD (63%), worked at an academic institution (97%) in Europe (58%), and predominantly studied basic biological mechanisms (68%). A large proportion of the participants (71%) had received some type of formal statistical training but 41% were not, or not very, familiar with factorial designs. Whilst 63% of participants are always or often, involved, or can influence, the planning of experiments, the majority (62%) have only incorporated females and males in 50% or less of their studies. On average, the participants had been involved in animal research for 13.8 years.

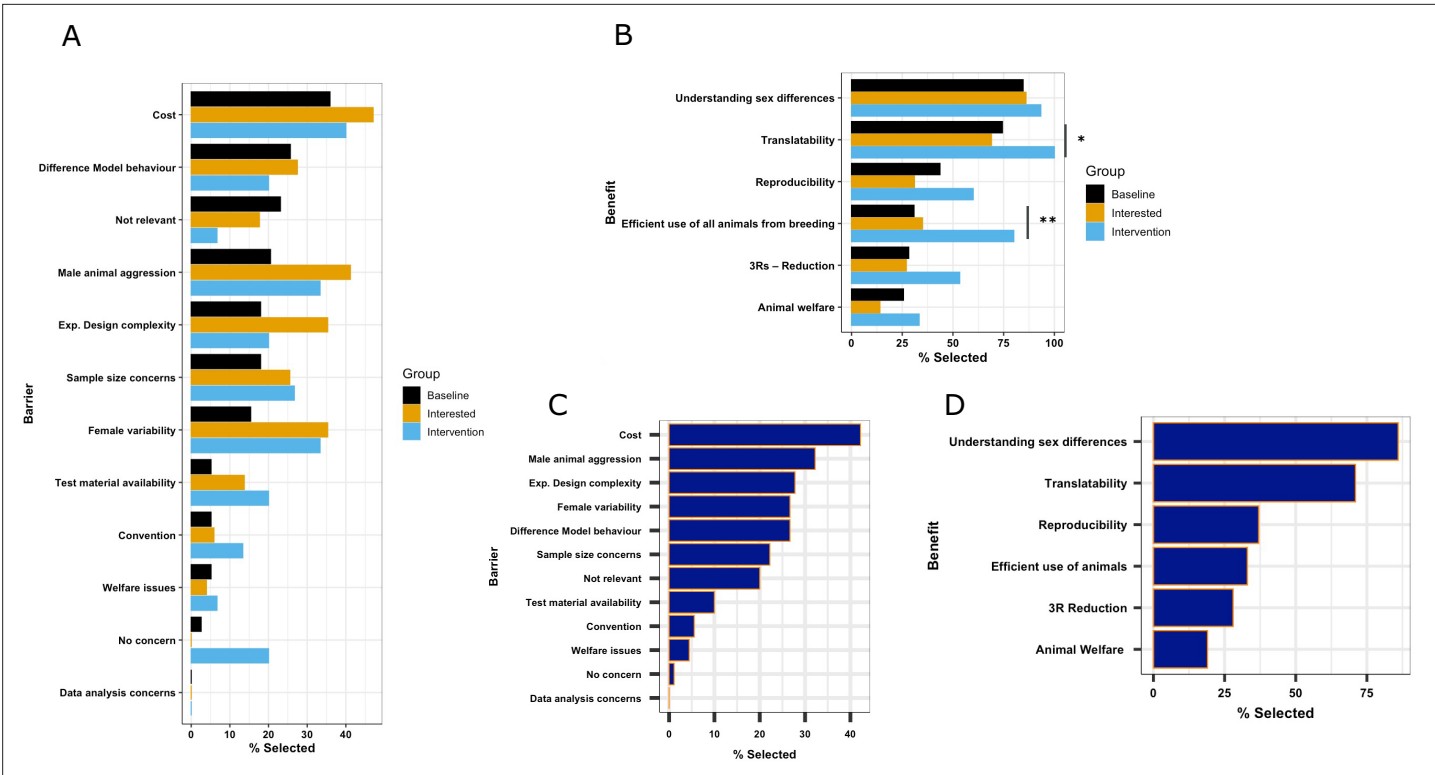

**Figure 1.** Exploration of the perceived barriers and benefits of inclusion of males and females in in vivo research from study 1. Evaluation of survey data collected in study 1 with 39 participants in the baseline group, 51 in the interested group, and 15 in the intervention group. (**A**) Percentage of study participants who selected each barrier, displayed by treatment group. (**B**) Percentage of study participants who selected each benefit, displayed by treatment group. (**C**) Percentage of study participants who selected each barrier in the general population (baseline and interested group combined). (**D**) Percentage of study participants who selected each benefit in the general population (baseline and interested group combined). A Pearson's chi-squared test was used to compare the proportion of participants selecting each barrier/benefit between the treatment groups. Statistical significance is highlighted with a horizontal bar and if the *p*-value is less than 0.05, it is flagged with one star (*). If the *p*-value is less than 0.01, it is flagged with 2 stars (**). If a *p*-value is less than 0.001 it is flagged with 3 stars (***).

## Study 2

Analysis of the demographic data (**Supplementary file 2**) found that the majority of the participants identified as female (52%), had an average age of 35 years, had a PhD (68%), and predominantly studied basic biological mechanisms (52%). A large proportion of the participants had received some type of formal statistical training (68%) but 55% were not, or not very, familiar with factorial designs. Whilst 81% of participants are always, or often, involved, or can influence, the planning of experimental designs, the majority (51%) have only incorporated females and males in 50% or less of their studies. On average, the participants had been involved with animal research for 10.6 years. No statistically significant correlation (*p*>0.05) was found between the continuous variables as assessed by a Pearson correlation analysis (**Appendix 1—table 2**).

## Exploration of advantages and barriers
### Study 1

Participants had an opportunity to articulate the barriers and benefits to inclusive research by either selecting preset options or entering free-text. For most preset options, there was no statistically significant difference seen in the proportion selected between the treatment groups (**Figure 1A–B**, **Appendix 2—table 1**, **Appendix 3—table 1**). We did not expect differences between the baseline and interested group but hypothesised that we would observe a shift in the intervention group following the workshop, with a reduction in those identifying misconceptions as barriers and an increase in the advantage statements selected. However, power is low to see a change in this exploratory proportion data due to the low sample size of eligible attendees at the workshop. Two exceptions occurred in

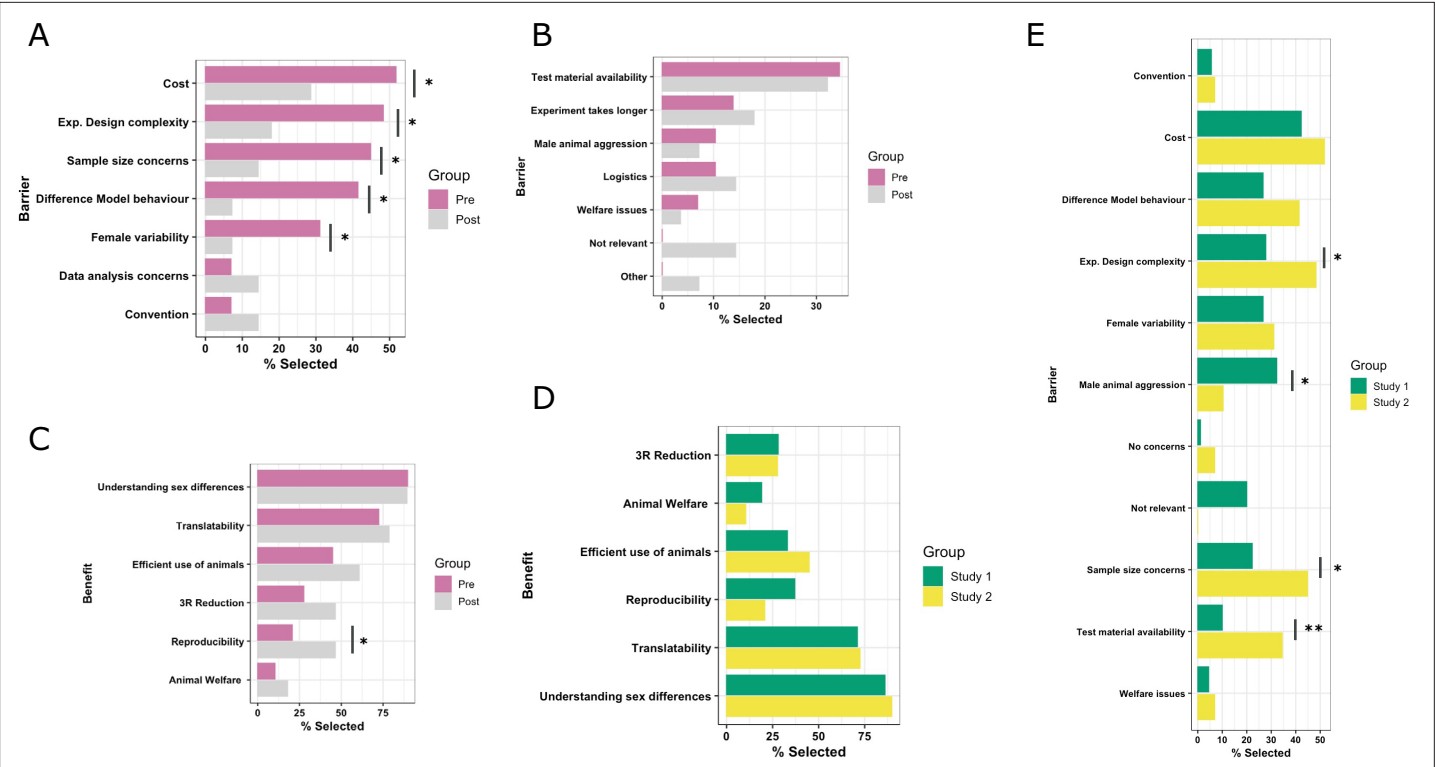

**Figure 2.** Exploration of the perceived barriers and benefits on the topic of inclusion of females and males in in vivo research for study 2. Evaluation of data collected from study 2 with 29 participants completing the barrier question pre-intervention and 28 in the post-intervention. (**A**) Percentage of study participants who selected each barrier, pre- and post-intervention, for the barriers targeted by the intervention. (**B**) Percentage of study participants who selected all other barriers, displayed pre-and post-intervention. (**C**) Percentage of study participants that selected each benefit, displayed pre- and post-intervention. (**D**) Comparison of the percentage of study participants selecting benefits in study 1 general population (baseline plus interested group) versus study 2 pre-intervention group. (**E**) Comparison of the percentage of study participants selecting barriers in study 1 general population (baseline plus interested groups) versus study 2 pre-intervention group. A McNemar's test of association was used to compare the proportions between the pre- and post-intervention data in study 2. A chi-squared test was used to compare proportions between the two studies. Statistical significance is highlighted with a horizontal bar, and if the *p*-value is less than 0.05, it is flagged with one star (*). If the *p*-value is less than 0.01, it is flagged with 2 stars (**). If a *p*-value is less than 0.001, it is flagged with 3 stars (***).

the benefit question. First, for the preset option: 'efficient use of all animals from breeding,' we saw a statistically significant increase for the intervention group selecting this as a benefit of inclusive research (34.2% and 37.5% selecting this option in the baseline and interested group, respectively, which then increased to 80% in the intervention group). We also observed a statistically significant increase for the intervention group selecting the translatability option (74% baseline, 68% interested, 100% intervention group).

To represent the general population, the baseline and interested treatment groups' data was pooled (*Figure 1C–D*). A higher proportion of participants selected benefits that are associated with generalisability (translatability and understanding sex differences) over robustness (reproducibility) and ethical use of animals. A small proportion of the sample (1.2%) did not perceive any barriers. The predominate barrier was associated with cost (48.9%). 35% of the participants raised male animal aggression. Barriers associated with misconceptions (female variability and sample size concerns) were raised by 25% of the participants.

## Study 2

The study design allowed a comparison of the pre- versus post-selection of barriers and benefits. For some barriers, statistically significant differences were seen in the proportion of time they were selected (*Appendix 4—table 1*, *Figure 2E*). The workshop was designed to challenge seven of the most frequently cited barriers by highlighting that they were based on misconceptions. The intervention produced a statistically significant reduction in five out of the seven barriers, with an

**Table 1.** Workshop Intervention construct.

| Section | Content |
|---|---|
| Lecture | Terminology<br>Reflection that clinical sex matters.<br>Exploration of the status of sex inclusion in preclinical research.<br>Exploration of what is sex-inclusive research.<br>The role of factorial analysis in sex-inclusive research. |
| Interactive element | Multiple choice questions with voting and discussion on appropriate analysis strategy and understanding factorial output. |
| Lecture | Exploration of the perceived barriers to sex-inclusive research. |
| Evaluation tool | Introduction of the Sex Inclusive Research Framework (SIRF) which can be used to evaluate research proposals from a sex inclusive perspective. |
| Interactive element | Multiple choice classification using the SIRF to classify justifications given for one-sex designs. |

average reduction of 28% citing the barrier post-workshop (*Figure 2A*). Participants had the option of selecting alternative barriers and entering free-text options. Of these, there was no statistically significant change in the proportions selected between the data collected pre- and post-intervention (*Figure 2B*). For some of the barriers, the proportions observed in the general population in study 1 (baseline and interested community) were similar to those seen in study 2 in pre-intervention data (e.g. 'Cost,' 'Female variability'; *Figure 2E*), whilst other barriers saw a statistically significant difference between studies, which could represent a difference in the local culture and practices. For example, relative to study 1, there was a 24% increase in the proportion selecting test material availability in study 2, whilst we observed a 25% reduction in those selecting male animal aggression concerns in study 2.

A comparison of the pre-versus post-data (*Figure 2C*) found the intervention had little impact on the proportion of time benefits were selected, except for 'reproducibility', where the intervention led to a statistically significant increase (26%). The proportion of time benefits selected were equivalent between study 1 and study 2 (*Figure 2D*; *Appendix 5—table 1*, *Table 1*).

**Table 2.** Statistical model output for the full model for study 1 data exploring the predictors ability to explain variation in average intent.

| Source | Nparm | Df | Sum of Squares | F Ratio | Prob >F | Significance | Eta² |
|---|---|---|---|---|---|---|---|
| Attitude | 1 | 1 | 18.5650 | 34.9548 | <.0001 | *** | 0.15 |
| Beh_Control | 1 | 1 | 0.0814 | 0.1534 | 0.6964 | | 0.00 |
| Soc_Norm | 1 | 1 | 12.2229 | 23.0137 | <.0001 | *** | 0.10 |
| Treatment_Group | 2 | 2 | 3.8643 | 3.6380 | 0.0311 | * | 0.03 |
| Age | 1 | 1 | 0.0995 | 0.1874 | 0.6664 | | 0.00 |
| Gender | 2 | 2 | 2.6196 | 2.4662 | 0.0918 | | 0.02 |
| Geography | 3 | 3 | 0.1955 | 0.1227 | 0.9464 | | 0.00 |
| Year_Work | 1 | 1 | 0.1828 | 0.3442 | 0.5592 | | 0.00 |
| Type_Work | 3 | 3 | 1.1958 | 0.7505 | 0.5255 | | 0.01 |
| Education | 2 | 2 | 2.9339 | 2.7621 | 0.0696 | | 0.02 |
| Stats_Training | 2 | 2 | 0.3810 | 0.3587 | 0.6998 | | 0.00 |
| Factorial_Fam | 1 | 1 | 0.0332 | 0.0625 | 0.8033 | | 0.00 |
| Factorial_Incor | 1 | 1 | 0.2614 | 0.4921 | 0.4852 | | 0.00 |
| Ability_Influence | 1 | 1 | 2.3253 | 4.3782 | 0.0398 | * | 0.02 |

Where Beh_control represents the average behavioural control score, Soc_norm the average social normal score, Year_Work represented the number of years the participants have worked in animal research, Type_Work represents the type of research conducted by the participant, Education the highest level of education obtained, Stats_Training represents the level of statistical training received, Factorial_Fam represents how familiar the participants were with factorial experimental design, Factorial_Incor represents how often the participants incorporated males and females into their experiments while studying an intervention, attitude represents the average attitude score, and Ability_Influence represents how often the participants were involved or could influence the planning of experiments involving animals. Nparm stands for the number of parameters, Df represents the degrees of freedom, and Prob >F represents the p-value associated with the F ratio. Statistical significance shown as * for *p*-value<0.05, ** for *p*<0.01 and *** for *p*<0.001.

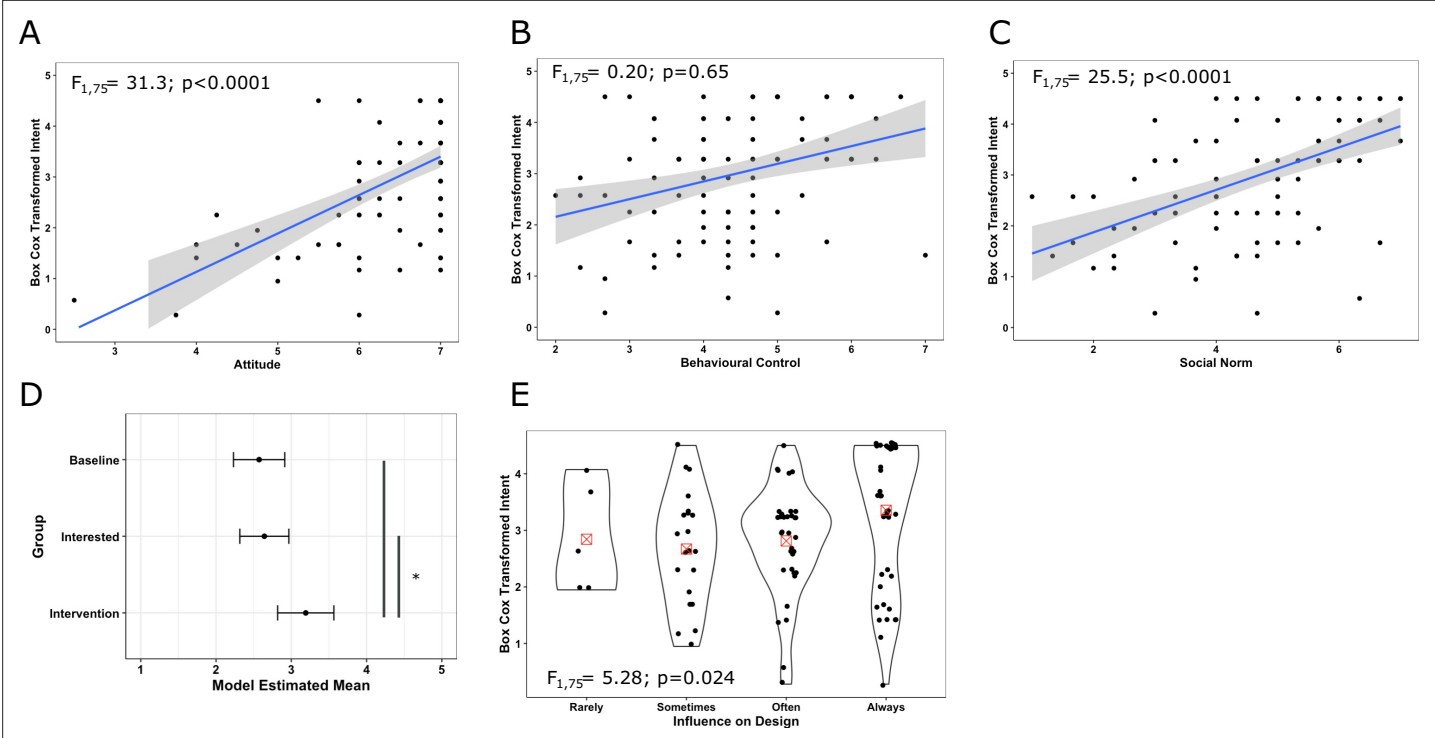

**Figure 3.** Exploration of intent for significant and critical predictor variables for study 1 data. A full model, with all potential predictors and demographics, was fitted to explore the variation in intent and assess for evidence of predictive behaviour (N=39 Baseline group, 51 Interested group and 15 intervention group). The baseline group was set as the reference group. If main effects were significant, the variation by treatment group was explored with Tukey post hoc testing. (**A**) Relationship between intention and attitude. (**B**) Relationship between intention and behavioural control. (**C**) Relationship between intention and social norm. For panels **A**, **B**, and **C**, the grey area indicates the 95% confidence interval for the fitted linear relationship (blue) and the text indicates the statistical significance of the relationship. (**D**) Model estimated means (Least Square Means) for each treatment group with a standard error bar estimated from the model. Vertical bars represent the planned comparison between groups, with * representing statistical significance <0.05. (**E**) Violin plot showing the distribution of intent as a function of the ability to influence the design. Points indicate individual study participants, and the red box indicates the calculated mean for each group. The text indicates the statistical assessment for that variable.

## Exploration of intention
### Study 1

The average intention score across all treatment groups for study 1 was high (5.67 ± 1.19, possible range 1:7). A statistical analysis assessed which variables could explain variation in intent to conduct sex-inclusive research (*Table 2*, *Figure 3*). Two of the three theories of planned behaviour attributes, subjective norm and attitude, significantly positively correlated with intent (*Figure 3A–C*). The data indicates that as the participant's attitude increases (i.e. feel incorporating males and females is important) their intention to incorporate females and males increases, with a high mean attitude of 6.5 ± 0.87. The subjective norm attribute had a similar positive correlation with intent, indicating that when participants perceived higher societal pressure, they had higher intention. Subjective norm differed from attitude in that, at the population level, the score was in the mid-range (4.7 ± 1.4). The intent for those who attended the workshop was significantly higher, with approximately a 20% increase in the intention score (*Figure 3D*). However, no significant difference in intent was found between the baseline and interested treatment groups. The only other variable to have some evidence of predictive ability was how often a participant was involved in or could influence the planning of experiments, which positively correlated with intent to incorporate females and males (*Table 2*, *Figure 3E*). Thus, suggesting individuals more involved in planning experiments had a higher intention to run inclusive designs.

**Table 3.** Statistical model output for the full model for study 2 data exploring the predictors ability to explain variation in average intent.

| Source | Nparm | Df | Df Denominator | F Ratio | Prob >F | Significance | Eta² |
|---|---|---|---|---|---|---|---|
| Attitude | 1 | 1 | 38.78 | 0.0268 | 0.8708 | | 6.91e-04 |
| Beh_Control | 1 | 1 | 42.47 | 1.9080 | 0.1744 | | 0.04 |
| Soc_Norm | 1 | 1 | 29.15 | 19.658 | **0.0001** | *** | 0.40 |
| Intervention | 1 | 1 | 40.06 | 0.0758 | 0.7845 | | 1.89e-03 |
| Age | 1 | 1 | 28.2 | 5.3428 | **0.0283** | * | 0.16 |
| Gender | 2 | 2 | 33.93 | 0.2973 | 0.7447 | | 0.02 |
| Ability_Influence | 1 | 1 | 27.91 | 2.2618 | 0.1438 | | 0.07 |
| Factorial_Fam | 1 | 1 | 42.98 | 0.3822 | 0.5397 | | 8.81e-03 |
| Factorial_Incor | 1 | 1 | 35.72 | 0.0725 | 0.7893 | | 2.03e-03 |

Where Beh_control represent the average behavioural control score, Soc_norm the average social normal score, Year_Work represented the number of years the participants have worked in animal research, Type_Work represents the type of research conducted by the participant, Education the highest level of education obtained, Stats_Training represents the level of statistical training received, Factorial_Fam represents how familiar the participants were with factorial experimental design, Factorial_Incor represents how often the participants incorporated males and females into their experiments while studying an intervention, attitude represents the average attitude score, and Ability_Influence represents how often the participants were involved or could influence the planning of experiments involving animals. Nparm stands for the number of parameters, Df represents the degrees of freedom, and Prob > F represent the p-value associated with the F ratio. Statistical significance is flagged with one star (*) if the p-value is less than 0.05, with 2 stars (**) if less than 0.01, and with 3 stars (***) if less than 0.001.

## Study 2

In an exploration of intention for study 2, only subjective norm positively correlated with intent to conduct a sex-inclusive study (*Table 3*, *Figure 4A–C*). Attitude did not predict intention but was found to have a very high average score (6.7 ± 0.51). Like study 1, behavioural control was not predictive but had a mid-range average score (4.3 ± 1.14). The workshop intervention was not found to explain variation in the intention score (*Table 3*, *Figure 4D*). However, the intention score in the pre-test group (5.71 ± 1.08) was only slightly higher than the intention score in the general population in study 1 (baseline and general interest; 5.51 ± 1.2). Age was also found to be negatively correlated, indicating older staff were less engaged with sex-inclusive research in this dataset (*Table 3*, *Figure 4E*).

## Exploration of knowledge

### Study 1

To complement the evaluation of intention and barriers and advantages, five questions were included to assess the knowledge of the participants on common misconceptions and errors around planning and data analysis of sex-inclusive studies (*Appendix 6—table 1*, *Figure 5*). The cumulative knowledge score (*Supplementary file 9*) did not significantly differ between the baseline and interested treatment groups ($z = -0.513$; $p=0.601$) (*Figure 5A*). Whilst the workshop intervention group had a significantly higher score relative to the baseline ($z=3.515$; $p=0.0044$) with an average increase of 1.94 questions correctly answered. This could also be seen in a comparison of proportion of correct answers for each individual question with a statistically significant higher score in the intervention group for 3 out of the 5 questions (*Figure 5B*).

### Study 2

The knowledge questions allow exploration in dataset 2 on the common misconceptions around sex-inclusive designs and common errors in data analysis of sex-inclusive studies (*Appendix 7—table 1*, *Figure 5*). The workshop intervention was found to significantly improve the score ($t=4.328$, df = 25, $p=0.0002$) with an average increase of 1.769 questions answered correctly (*Supplementary file 10*, *Figure 5C*), a similar effect size to study 1. The impact of the intervention could also be seen at an individual question level with a statistically significant higher score in the intervention group for four out of the five questions (*Figure 5D*).

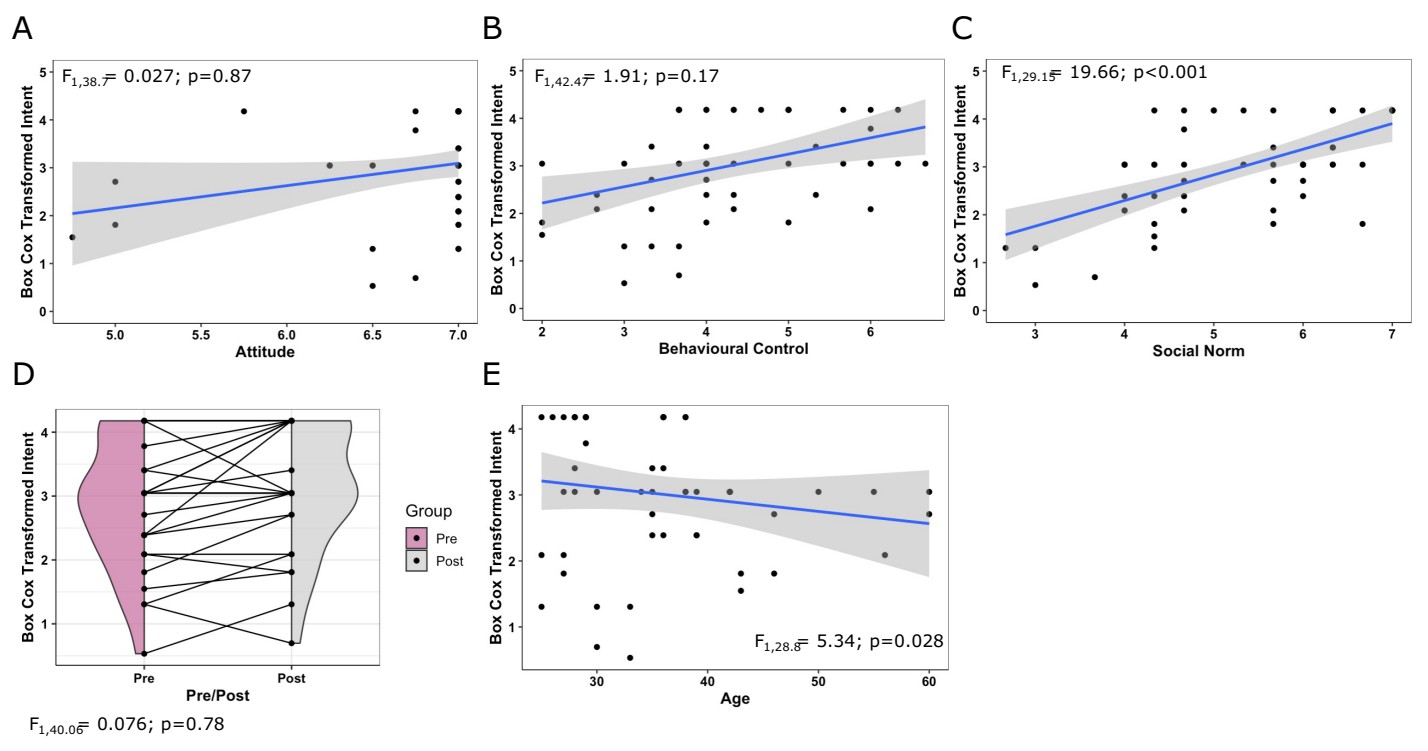

**Figure 4.** Exploration of intent for significant and critical predictor variables for study 2 data. A full model, with all potential predictors and demographics, was fitted to explore the variation in intent and assess for evidence of predictive behaviour. (**A**) Relationship between intention and attitude. (**B**) Relationship between intention and behavioural control. (**C**) Relationship between intention and social norm. For panels **A**, **B**, and **C**, the grey area indicates the 95% confidence interval for the fitted linear relationship (blue) and the text indicates the statistical significance of the relationship. (**D**) Exploration of intention between pre- and post-intervention where a line links each individual participants score and shaded area indicates the density of the Box-Cox transformed measure of intent. (**E**) Relationship between intention and age. For graphs **A**, **B**, **C**, and **E**, the grey area indicates the 95% confidence interval for the fitted linear relationship (blue) for Box-Cox transformed intent (y-axis) against key predictors in the TPB model (x-axis) (**A**, **B**, **C**) and significant predictor 'age' (**E**).

## Discussion

### General population conclusions – barriers/benefits/knowledge

The survey provided an opportunity to understand researchers' position around sex-inclusive research. Across the two studies, the benefits of sex-inclusive research were selected at a similar rate with a higher proportion of participants valuing the benefits associated with generalisability of the findings (ability to understand sex differences and translation) over robust ethical research. This recognition of the benefit of inclusive research aligns with previous suggestions that scientists are generally supportive of efforts to improve sex representation in biomedical research (*Karp and Reavey, 2019*; *Medical Research Council Working Group, 2022a*). However, the focus on the ability to understand sex differences aligns with a common misconception that the goal of inclusion is to identify sex differences. This misconception causes a dissonance with the recommendation to share the sample size (N) for a treatment effect between the two sexes, as this approach does not power experiments to identify sex differences. Being explicit on the experimental design goals and the impact on the design was raised by *Rich-Edwards and Maney, 2023* who developed a '4 Cs' decision framework to guide researchers in sex-inclusive research by focusing on four steps: Consider, Collect, Characterise, and Communicate. Within this framework, they identify two pathways depending on whether the study is an exploratory study (where males and females are included to improve generalizability) versus confirmatory study (those actively exploring sex-related differences) and guide researchers in the design and analysis implication of this. There is a culture shift needed that the primary goal in exploratory sex inclusion is to deliver generalisable results where large differences in sex-related variation will be identified. It is this understanding of the difference between exploratory versus confirmatory sex-inclusive research that, in our opinion, is deficient.

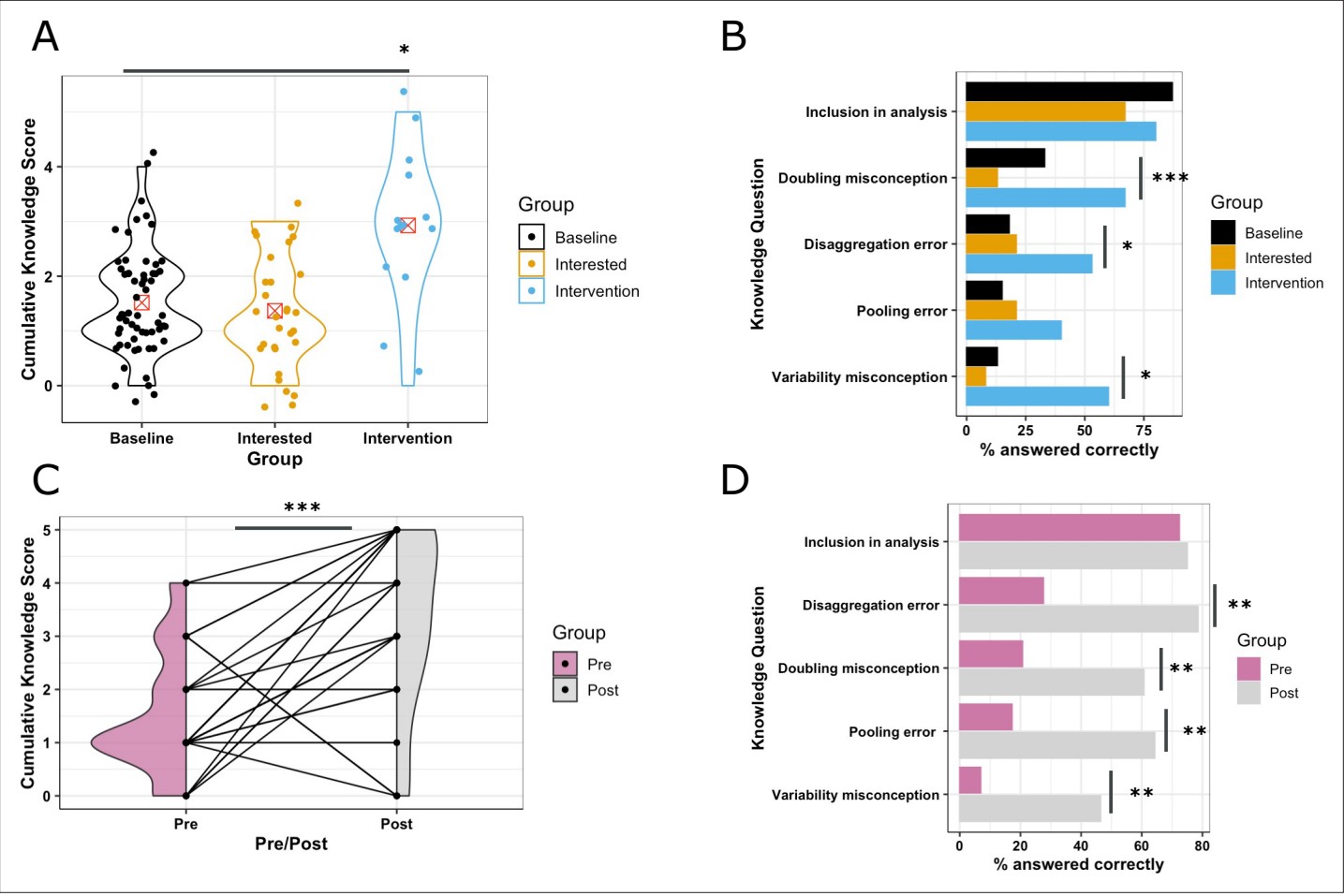

**Figure 5.** Exploration of intervention impact on the proportion of correctly answered knowledge questions. (**A**) Study 1: Cumulative knowledge score (cumulative questions answered correctly) displayed by treatment group (Baseline: N=39, interest: N=51 and intervention: N=15). Statistical significance assessed with a Poisson regression. (**B**) Study 1: Percentage of correct answers for each question, displayed by treatment group (Baseline: N=39, interest: N=51, and intervention: N=15). Statistical significance assessed with a Pearson's chi-squared test. (**C**) Study 2: Impact of intervention on the cumulative knowledge score. Where a line links each individual participants score, and shaded area indicates the density of the Box-Cox transformed measure of intent. Statistical significance assessed with a paired *t*-test (N=26 with pre and post data available). (**D**) Study 2: Percentage correct answers for each question, displayed by pre- and post-intervention group. (Pre: N=29, Post: N=28). Statistical significance assessed with McNemar's test of association. Statistical significance shown as * for *p*-value<0.05, ** for *p*<0.01 and *** for *p*<0.001.

Research has found that scientists do not believe that inclusion is 'doable' (*Gompers, 2018*) due to a range of perceived barriers to inclusion (*Medical Research Council, 2022*; *Phillips et al., 2023*; *Karp and Reavey, 2019*; *Prendergast et al., 2014*; *UK Research and Innovation (UKRI), 2023*). This study provides the first quantification of the scale of these perceived barriers, of which many are misconceptions, and finds them endemic. Seldom did the sampled community express the absence of barriers. Cost was the most cited barrier, approaching half of the community raising this as a concern. This was followed by, on average, a third of the community selecting barriers related to misconceptions such as experimental design complexity, sample size concerns, and female variability.

For some of the barriers, we observed a statistically significant difference between studies (test material availability and male animal aggression), which could represent a difference in the local culture or research interests impacting the perceived barriers. For example: In study 1, male mouse aggression was raised by 30% of the respondents but only 10% in study 2. This finding is at odds with the observation that male animal research predominates (*Woitowich et al., 2020*) but could relate to welfare management practices prioritised within that community. Sharing best practices and working on housing and husbandry conditions could minimise the impact of this concern (*Van Loo et al., 2003*; *Blankenberger et al., 2018*; *Jirkof et al., 2020*).

The responses to the knowledge questions also give an opportunity to directly assess whether participants are holding a misconception. By focusing on the baseline and interested group in study 1 or the pre-intervention data in study 2, we get an estimate of the general research population. This finds that misconceptions dominate the research community beliefs. For example, the misconception that inclusion of males and females would double the N (80% of study 1 N=90, 79% of study 2 N=28) held this view. This is probably arising from the perception that sex-inclusive designs increased variability and, therefore, require a larger overall sample size (90% of study 1 N=90, 93% of study 2 N=28). This perception has been acknowledged in previously published articles (*Medical Research Council, 2022*; *Waltz et al., 2021*; *Gompers, 2018*; *Prendergast et al., 2014*; *Beery, 2018*), however, this is the first known quantification of the extent of these viewpoints.

Regarding the data analysis, most researchers (75.6% in study 1 general population N=90, and 72% of the participants in pre-assessment of study 2 N=28) correctly knew that sex should be included in the statistical model. Questions that are more specific about analysis strategies find the proportion of incorrect answers to be very high. For example, most thought the data could be pooled across the sexes (81.1% in study 1 general population, 83% study 2 pre-assessment) and that the data should be disaggregated (80% in study 1 general population, 72% study 2 pre-assessment). The prevalence of incorrect ideas around data analysis aligns with a published meta-analysis that analysis mistakes are common when females and males are included in addition to the intervention of interest in research (*Garcia-Sifuentes and Maney, 2021*). The high proportion of correct answers for the initial question to include sex but then high proportion of agreeing with poor analytical strategies suggests that participants have a sense they need to do something but either don't know what the right analysis strategy is or are unaware of the problems with the suboptimal analysis strategies.

It is of note that no one selected data analysis concerns in study 1 and infrequently in study 2 (7% in the pre-test assessment) as a barrier to inclusion which suggests researchers are unaware of this shortcoming. This aligns with research identifying that statistical errors in the analysis of sex-inclusive datasets are common (*Garcia-Sifuentes and Maney, 2021*). The difficulty in recognizing one's own incompetence is a known cognitive bias, described as the Dunning-Kruger effect (*Kruger and Dunning, 1999*). Research has shown that training not only improves participants skills but also helps the individuals recognize the limitations of their ability (*Kruger and Dunning, 1999*). This highlights the importance of investing in training researchers in experimental design and data analysis and that this is a critical step to enable sex-inclusive research.

These findings highlight the scale of the misconceptions hindering engagement with sex-inclusive research and the need for research practice leaders to focus on both the imbedded these general misconceptions but also to determine local barriers that need exploration with their research communities.

## Exploration of intention to inclusive research

While training is crucially important, it alone is unlikely to completely solve the embedded cultural problem we currently face. Simply knowing that one should exercise to improve your health, does not mean one will get up every morning to go to the gym. Thus, knowing exactly what drivers are needed to influence the required behavioural change are equally as important as making sure people receive training to enact the change. The theory of planned behaviour provides us with key insights on how to influence intention and ultimately behavioural change. This is key as the intention to conduct sex-inclusive research in the future was generally positive (5.51 ± 1.25 in study 1 general population, 5.71 ± 1.08 study 2 pre-assessment out of a total score of 7) and attitude was very high (6.41 ± 0.92 in study 1 general population, 6.66 ± 0.67 study 2 pre-assessment out of a score of 7). This indicates that scientists generally view sex inclusive research as beneficial, useful, and the right thing to do. Despite this, when asked how often in the past 5 years participants had included female and male samples in their experiments, 35% (general population) in study 1 and 55% in study 2 (pre-assessment) indicated almost always or often (51–100% of the time) incorporating male and female samples in their studies. Thus, confirming the above posit that simply knowing sex-inclusive research is important is not enough to influence active incorporation, and since attitude is already high, it leaves little room for further influence on behavioural intent.

Although attitude was high, subjective norm differed at the population level where the score was more in the mid-range (4.6 ± 1.4 in study 1 general population, 5.14 ± 1.11 study 2 pre-assessment).

This factor was also the only one of the three theories of planned behaviour's beliefs found to significantly influence intent in both studies with large effect sizes. Understanding the strong positive correlation between subjective norm and intent allows for focused interventions to increase societal pressure on the scientific community which should then improve overall intention towards sex-inclusive research.

While researchers may have the right motivation, performing the behaviour is dependent on other non-motivational factors such as the time, money, or skills (*Ajzen, 1991*). A behaviour can only be conducted if the researcher is the decision maker. Not only does actual control influence behaviour but so does the perception of behavioural control, such as self-efficacy. Self-efficacy encompasses both how well a person can do the behaviour but also how confident they feel about doing it. Self-efficacy opinions can influence choice of activities, preparation, as well as effort. In this case, we believe that the lack of significance in our model is due to the false perception of how to accurately design and analyse factorial designs, evidenced by the low knowledge score. According to Ajzen, perceived behavioural control may not be realistic if a participant has little knowledge about the behaviour in question (*Ajzen, 1991*). Again, based on the low knowledge scores of how to design and analyse sex-inclusive data may indicate that this measure may not add much accuracy to behavioural prediction. However, it does indicate the overwhelming need to create a training that is accessible and practical to all in vivo scientists.

## Impact of the intervention on intention

We had an opportunity to incorporate a workshop at an international conference and as part of this, we decided to assess how effective that training was. As many have experienced, the practical aspects and decisions that need to be made in an in vivo experiment are rarely covered in traditional statistics courses. This often leaves scientists unsure how to apply what they have learned in the real world. Furthermore, 26–29% of the participants in this study had received no formal statistical training, and yet 71–75% can always or often influence the experimental design, illustrating the need for continuing education and practical instruction for established scientists. In the first study, trained attendees had the highest level of intent to conduct sex-inclusive research in the future, potentially indicating that our workshop was effective. However, it is also plausible that those who signed up inherently had a higher level of intent. Thus, we felt it was important to evaluate attendees' intent before and after the training. This naturally led to our second study, where we found that intent was not altered by the workshop, because it was already high. Even though intent was not statistically altered (Pre: 5.71 ± 1.08; Post: 6.00 ± 1.03), participants knowledge increased and fewer barriers were selected after the training. This illustrates that education alone may not be enough to change behaviour but utilizing the insight gained that social pressure may be the best avenue to influence actual behavioural change.

## Impact of the intervention on perceived benefits and barriers

The primary goal of the intervention was to challenge misconceptions; however, the workshop material naturally started the learning journey by exploring the drivers for, and barriers preventing, change. When focused on the benefits, we found that following the training intervention, participants tended to select each of the benefits at a higher proportion compared to other groups. There was a statistically significant change in a few of the benefits: a large increase in the selection of efficient use of animals and translatability in study 1 and reproducibility in study 2. The infrequent statistical significance of the change at the individual benefit level is likely to arise from low statistical power due to the high baseline signal and low sample size.

An exploration of the study 2 data found that the intervention did not generally reduce the selection of barriers but did decrease the selection of the barriers associated with the misconceptions. This demonstrates that the targeted intervention was effective, with a meaningful effect size (a 28% reduction on average), at improving the perception that generalisable sex-inclusive research is feasible.

## Impact of intervention on knowledge

It has been highlighted previously that early career researchers lack awareness of best practice recommendations across their areas of research (*Morahan et al., 2024*). To address this, it is important that training initiatives directed at improving knowledge of best practice are put in place by institutions, with the goal of continuously improving practice and supporting cultural change. This means that

training must extend beyond that provided simply as part of graduate and post-graduate studies, and that implementing training opportunities is part of continuing professional development of research staff.

In the baseline and pre-intervention data, performance on the panel of knowledge questions was generally poor, with only the question around whether sex should be included in analysis answered correctly by more than half the participants. This is consistent with previous findings that statistical approaches, even studies designed as sex inclusive, are frequently erroneous (*Garcia-Sifuentes and Maney, 2021*). Encouragingly, post-intervention participants demonstrated significant improvements in knowledge. For example, far more post-intervention researchers correctly identified that disaggregation and/or pooling of the sexes within a statistical analysis was a mistake, particularly in study 2.

Given that the majority of commonly cited barriers stem from misconceptions which are arising from knowledge gaps, the evidence that targeted interventions can directly enhance knowledge offers promise for promoting cultural change through educational initiatives. Despite across-the-board improvements in performance, the outcomes still leave room for further improvement. For example, nearly 40% of post-intervention participants still expressed the view that inclusion of female and male samples required a doubling of the sample size. This highlights that a single session is likely insufficient to equip researchers with all relevant knowledge and would thus support a future strategy of continuous improvement through multiple training and learning strategies.

## Limitations of the study

Whilst the data indicate a clear benefit of the workshop intervention, there are several caveats and limitations that should be considered and acknowledged. First and foremost, we did not directly measure the behaviour of interest, and it is possible that participants may have over-exaggerated their current and future use of sex-inclusive designs. The basis of the theory is that people are more likely to perform a behaviour if they plan to do so (intentions) (*Ajzen, 1991*) and has been shown to be highly predictive. Therefore, measuring intention can still provide valuable insight into the perspectives of scientists and how their beliefs may impact implementation of sex-inclusive designs. For both studies, university researchers made up the majority of the population. The workshop was hosted by a recognised senior leader (Professor Ahluwalia) both at the World Congress (role: WCP2023 Secretary-General) and within the university setting (role: Dean for research at the Faculty of Medicine and Dentistry). It is possible that participant awareness of the views of the senior leader may have influenced the positive outcome of the intervention. For study 1, the congress was specifically focused for pharmacologists, and it is possible that the outcomes reflect only this sector of biomedical research. However, for study 2, the workshop was advertised across a multi-faculty university and likely represented a more specialty-diverse population. Whether the workshop would effect a similar change in understanding from those who work within the pharmaceutical industry requires further evaluation. The impact of the workshop could depend on the skills and experience of the trainers. Furthermore, the results could be specific to a face-to-face training event and whether a virtual course (which has the benefit of scale) would have a similar benefit would need further exploration. This workshop focused on the barriers and misconceptions around the design and analysis. The material did not tackle any of the practical barriers e.g. management of welfare issues with male mice. There will be a need for future work to explore how to support and address these barriers. Finally, the questions posed to workshop attendees did not state whether the queries were being posed for a first exploratory study or not. It is possible that had this been made explicit then the responses could have differed.

## Conclusions

An endemic and persistent sex bias in early research has been raised as a risk to the validity of biological knowledge. Sociological research finds that scientists believe sex matters, but change has been limited. Funders are actively trying to change behaviour by requiring a justification for exclusion. Changing the status quo requires individuals to buy into a new direction that aligns with their values and beliefs. Here, we demonstrate that a workshop intervention can rescue the core beliefs that hinder sex inclusive research. We also provide evidence that institutes, funders, and professional bodies can help this journey by raising awareness of best practice, as this will change the perceived cultural norms. All resources used in this research (both the teaching material and the surveys) have been made freely available as supplementary material to support further research and activities.

## Materials and methods

All procedures and informed consent protocols were approved by the Queen Mary University of London Ethics Research Committee (ID: QMERC23.001). The protocols obtained informed consent after the nature and possible consequences of the study were explained. In November 2023, an amendment was approved to run additional workshops for staff and students at Queen Mary University of London (ID: QMERC23.002).

### Study 1 experimental design

In this study, data from three treatment groups (baseline, interested, intervention) were collected at the 19th World Congress of Basic & Clinical Pharmacology 2023 in Glasgow, Scotland with an online questionnaire (*Supplementary file 5*) by convenience sampling within each treatment group. Participants attending the poster sessions on 3rd and 4th July were recruited by circulating staff to contribute to the baseline treatment group, thus sampling the general population that attended the conference. Participants for the interested group were recruited by approaching attendees who were in attendance at a related symposium advertised in the conference proceedings on the 4 July 2023 (entitled: 'The importance of interrogating sex differences in cardiovascular physiology and disease'). Participants for the intervention group were recruited by advertising a free educational workshop in the conference proceedings entitled 'Best practice for sex inclusive research' conducted on the 5 July 2023 and data were collected post-intervention. Workshop participants were given the option to participate in the survey at the end of the workshop. To compensate participants in any of the treatment groups for their time, they received an entry into a drawing for one of ten £50 Amazon e-gift cards. Participants were included if they conducted research on a disease or biological phenomenon that affects males and females and were able to influence the plan or conduct of in vivo research. At the point of data collection, the organisers were not masked (blinded) as we knew the hypothesis and the intervention group in the study. However, data was collected by sharing a QR code with the participants who then completed the data entry independently and, therefore, our ability to influence the results were minimal.

A sample size estimate was conducted a priori using the highest standard deviation (1.4) observed in a similar study (*LaFollette et al., 2020*) that also used a survey based on the theory of planned behaviour. With this standard deviation, to detect a change of one point between the groups, a power calculation based on the Student's *t*-test estimated 31 participants per group (62 total) would be needed to achieve a power of 0.8. We aimed to collect data from 50 participants in each treatment group, to ensure we had sufficient data after dropout (failure to attend or meet the inclusion criteria).

### Study 2 experimental design

In this study, a pre- and post-intervention design was implemented to assess the impact of the workshop with an online questionnaire (*Supplementary file 6* (pre-questions) and *Supplementary file 7* (post-questions)) with convenience sampling of a pre-existing group. Participants from Queen Mary University of London were recruited by advertising an educational workshop entitled 'Best Practice for Sex-Inclusive Research' conducted on the 14th of February 2024. To compensate participants for their time, they received an entry into a drawing for a single £50 Amazon e-gift card. Participants were included if they conducted research on a disease or biological phenomenon that affects females and males and were able to influence the plan or conduct of in vivo research.

We aimed to recruit 50 participants to the workshop and only collected data during the workshop. This was the maximum number we felt would be feasible to maintain the interactive nature of the workshop. The implementation of a pre- and post-design should have higher power than the previous study (see power calculation section 2.1) to detect changes. At the point of data collection, the organisers were not masked (blinded) as we knew the hypothesis and the intervention point in the study. However, data was collected by sharing a QR code with the participants who then completed the data entry independently and, therefore, our ability to influence the results were minimal.

### Survey construct

The survey was developed through consulting with experts in survey methodology, behaviour theory, and sex-inclusive research design and analysis. When possible, validated instruments were used (i.e. questions based on the TBP). When validated instrumentation did not exist, new items were created,

reviewed by statistical experts, piloted, and revised as necessary. Two cycles of pilots were conducted with in vivo pharmacology researchers (N=9) from both academic and pharmaceutical settings. Survey 1 consisted of 39 questions and survey 2 had a maximum of 34 questions in a testing phase (*Supplementary file 5*). Demographic questions were omitted from the post-question format in Study 2 and data from the same individual were aligned between the pre- and post-questions through the 3+ initials inputted by the researcher (pre-survey: *Supplementary file 6* and post-survey: *Supplementary file 7*).

The initial four questions obtained consent and assessed whether participants met the inclusion criteria. If the participant did not grant consent or failed to meet the inclusion criteria on any question, the survey was terminated. Thereafter, all questions were presented to each participant. Participants were asked demographic questions which included age, gender, geographic region (study 1 only), and highest level of education. Several questions captured information about their work including type of institution (e.g. academic, contract research organisation; study 1 only), primary type of research (e.g. applied, basic, regulatory), and number of years working with laboratory animals. In addition, questions were included to capture data on potential explanatory variables. These included their ability to influence the construct of experiments, the amount of statistical education they had received, familiarity with factorial designs, how often they incorporate males and females into their experimental designs, their knowledge on how to analyse data when females and males were collected, and the impact of including males and females on statistical power. Finally, a list of pre-identified advantages and barriers were provided to participants to list perceived reasons why incorporating females and males was considered advantageous or challenging. Participants were provided with an empty text box to list any other advantages or barriers in addition to the list provided.

Motivations, outlined by the theory of planned behaviour, regarding sex-inclusive research were assessed using a series of thirteen questions. Participants answered four closed-ended quantitative questions about their behavioural attitudes (relative perception about the positive or negative valence of sex inclusive research), three questions on subjective norms (social and professional pressures to conduct sex inclusive research) and three questions on perceived behavioural control (general confidence/control over the ability to run sex inclusive studies). The perceived behavioural control variable is different from self-efficacy as it asks participants about external control factors, such as whether inclusive designs are under the control of the participant. One open-ended question was included at the end of survey to assess if there were any other thoughts or comments participants wanted to share on the topic. Finally, three questions were used to assess participants' future intent to implement sex-inclusive research.

The survey includes a question to determine whether participants are answering the question as part of the baseline/interested/intervention group (study 1) or pre-/post-assessment (study 2). Multiple participation was not anticipated due to the length of the survey and the time constraints for data collection.

The surveys were constructed to ensure that the only identifiable information was the optional input of an email address to support the draw of the Amazon vouchers. To maintain anonymity, once data was downloaded, an ID code was assigned, email addresses removed, and the email information only used for the draw. The final data was stored in an encrypted, password-protected file accessible only to the research team.

## Workshop construct

The workshop, or training intervention, was developed by statisticians who have published, presented internationally, and taught on this topic . One of the statisticians has a formal teaching qualification and a background in education. The training consisted of a 1 hr 45 min workshop with didactic, interactive activities and introduced a framework to evaluate research proposals from a sex-inclusive perspective (*Table 1*, *Supplementary file 8*). Prior to study 1, it was run as a pilot with a small group of scientists to test the material. Following the results of study 1, the workshop material was refined by amending the multiple-choice questions one and two to be more explicit in driving a discussion around whether data should be analysed by pooling, disaggregation or with a factorial test. Furthermore, multiple-choice question six was added to re-enforce learning that a baseline sex difference can be separated from an intervention effect.

## Statistics and reproducibility

### Participant inclusion

In study 1, a total of 194 participants started the survey. Of these, 105 met the inclusion criteria questions (N=39 baseline, N=51 interested, and N=15 intervention). While all 105 met the inclusion criteria, seven participants left the question about age blank. Therefore, the final dataset to evaluate intent included 98 participants (N=35 baseline, N=48 interested, and N=15 intervention).

In study 2, a total of 63 individuals registered for the workshop, 42 participants attended all, or part of the workshop, and survey data was collected from 31 individuals. However, due to some not meeting the inclusion criteria, or not completing one of the surveys, we received data from 28 individuals for the pre-survey and 28 for the post.

### Variable coding

For the theory of planned behaviour questions, each question had a scale from 1 to 7. An average score was calculated for each motivation (attitude, perceived behavioural control, and social norm) and for intent. This strategy required that each participant answer at least 50% of the questions for each motivation, otherwise, their data was discarded. All participants answered at least 50% of the theory of planned behaviour questions and, therefore, no participant's data were discarded based on these criteria.

To aid analysis, demographic categories with less than ten responses were combined with similar-themed options into larger categories. For example, the question that asked about previous statistical training, the categories of 'No training' and 'Primarily informal/practical' were collapsed into a category called 'No courses.' Missing data for categorical variables (gender, geography, etc) were coded as 'unknown'.

Within the survey, two (survey 2) to three (survey 1) questions had opportunities for participants to add a free-text answer in response to a question as an additional option provided to a question. Those questions were about the perceived barriers to or advantages of using males and females in preclinical research. For these questions, all free-text submitted were reviewed by the research team and categorised based on perceived theme. Generally, very few free responses (Survey 1: 12 for barrier question only; Survey 2: 6 for barrier question only) were provided (*Appendix 8—table 1* and *Appendix 8—table 2*).

Five knowledge questions were posed to the participants based on misconceptions (*Phillips et al., 2023*) concerning using females and males simultaneously in an experiment and variability in the female sex. With each question having a single correct answer. If a participant provided the correct answer, they were given a point, and the points were summed across the five questions. The knowledge question summary metric was not included in the statistical exploration of intent, as knowledge was anticipated to correlate with treatment group.

### Quantitative analysis

Throughout this research, a significance level of $p<0.05$ was considered statistically significant. Multiple testing correction was not applied as this is an exploratory analysis.

### Evaluation of intent

All data analyses were conducted in JMP 14.0.0 (SAS Institute Inc, Cary, NC, USA). SAS was used to conduct the theory of planned behaviour intent analyses (*Supplementary file 11* - analysis).

The dependent variable for quantitative analysis was the average intent (Avg_Intent) which was Box-Cox transformed to improve the distributional characteristics in study 1 only. As fixed effects of interest, the statistical model included 'Treatment_Group' for study 1 and 'Intervention' for study 2. In addition, three theories of planned behaviour attributes: average attitude (Attitude), average behavioural control score (Beh_Control) and average social norm (Soc_Norm) were included as fixed effects. Furthermore, potential explanatory variables (e.g. demographic variables) were included in the model as fixed effects. A full model containing all possible terms (*Equation 1* study 1 and *Equation 2* study 2) was used, as we had no strong prior information that we could use to select predictive variables. For the data from study 2, a mixed model analysis was used as we included a random term for participant to account for the repeat nature of the data. For both studies, the baseline group was set as the reference group. Since a study of this kind has not been conducted before and the

social factors (age, gender, previous statistical experience, etc.) are not reported as explaining/not explaining variability in data such as this, we left all potential explanatory variables in the model to provide data on demographic variables for future social science work of this kind. We evaluated the final model with and without these non-significant variables and the inclusion or exclusion, did not affect the results presented. Model diagnostics (e.g. independence of residuals, homogeneity of variance, and normality of residuals) were inspected and no concerns on the model quality were identified.

Eta squared for study 2 was estimated using the F_to_eta2 function in the effect size library with R4.3.1.

$$
\begin{aligned}
\text{Avg\_Intent} = {}& \text{Treatment\_Group} + \text{Attitude} + \text{Beh\_Control} + \text{Soc\_Norm} + \\
& \text{Age} + \text{Gender} + \text{Geography} + \text{Year\_Work} + \text{Education} + \text{Type\_Work} + \\
& \text{Stats\_Training} + \text{Factorial\_Fam} + \text{Factorial\_Incor} + \text{Ability\_Influence}
\end{aligned}
\tag{1}
$$

$$
\begin{aligned}
\text{Avg\_Intent} = {}& \text{Intervention} + \text{Attitude} + \text{Beh\_Control} + \text{Soc\_Norm} + \\
& \text{Age} + \text{Gender} + \text{Ability\_Influence} + \text{Factorial\_Fam} + \text{Factorial\_Incor} \\
& + (1 \mid \text{Participant})
\end{aligned}
\tag{2}
$$

Where Year_Work represented the number of year the participants have worked in animal research, Education the highest level of education obtained ('Doctoral,, 'Masters,', 'Other'), Type_Work represents the type of research, Training represents the level of statistical training received ('No courses', '1–2 courses', >2 courses'), Factorial_Fam represents how familiar the participants were with factorial experimental design, Factorial_Incor represents how often the participants incorporated males and females into their experiments while studying an intervention and Ability_Influence represents how often the participants were involved or could influence the planning of experiments involving animals. Participant was included as a random effect in *Equation 2* and is represented in the model as (1|Participant). For study 2, few demographics could be included in the analysis due to lack of variability and the fact that several were not asked at the post-intervention time point.

## Evaluation of knowledge

Using the statistical programming language, R Version 4.3.1, a Poisson regression analysis assessed the role of treatment group in explaining variation in the cumulative knowledge score for study 1 (data: *Supplementary file 3*, analysis: *Supplementary file 12*) and study 2 (data: *Supplementary file 4*). A variety of diagnostics (e.g. residual distribution, assessment of Cook's distance, Leverage, goodness-of-fit test) were generated to ensure the model was robust and appropriate for the data.

# Acknowledgements

The funders had no role in the design of the study; in the collection, analyses, or interpretation of data; in the writing of the manuscript, or in the decision to publish the results. This work was supported by grants for AA from The British Journal of Pharmacology (https://bpspubs.onlinelibrary.wiley.com/journal/14765381) and AstraZeneca (https://www.astrazeneca.com/).

# Additional information

### Competing interests

Brianna N Gaskill: is affiliated with Novartis Biomedical Research. The author has no other competing interests to declare. Benjamin Phillips, Natasha A Karp: has shareholdings in AstraZeneca. The author has no other competing interests to declare. The other authors declare that no competing interests exist.

## Funding

| Funder | Grant reference number | Author |
|---|---|---|
| AstraZeneca (United Kingdom) | Inclusive Research Intervention | Amrita Ahluwalia |
| British Journal of Pharmacology | Inclusive Research Intervention | Amrita Ahluwalia |

The funders had no role in study design, data collection and interpretation, or the decision to submit the work for publication.

## Author contributions

Brianna N Gaskill, Conceptualization, Formal analysis, Investigation, Writing – original draft, Writing – review and editing; Benjamin Phillips, Investigation, Visualization, Writing – review and editing; Jonathan Ho, Holly Rafferty, Oladele Olajide Onada, Andrew Rooney, Investigation; Amrita Ahluwalia, Funding acquisition, Investigation, Writing – original draft, Writing – review and editing; Natasha A Karp, Conceptualization, Formal analysis, Supervision, Funding acquisition, Investigation, Visualization, Writing – original draft, Project administration, Writing – review and editing

## Author ORCIDs

Brianna N Gaskill ⓘ https://orcid.org/0000-0002-1884-803X
Benjamin Phillips ⓘ https://orcid.org/0000-0001-9161-8903
Andrew Rooney ⓘ https://orcid.org/0000-0001-7188-2213
Amrita Ahluwalia ⓘ https://orcid.org/0000-0001-7626-6399
Natasha A Karp ⓘ https://orcid.org/0000-0002-8404-2907

## Ethics

All procedures and informed consent protocols were approved by the Queen Mary University of London Ethics Research Committee (ID: QMERC23.001). The protocols obtained informed consent after the nature and possible consequences of the study were explained. In November 2023, an amendment was approved to run additional workshops for staff and students at Queen Mary University of London (ID: QMERC23.002).

Reviewer #1 (Public review): https://doi.org/10.7554/eLife.106545.3.sa1
Reviewer #2 (Public review): https://doi.org/10.7554/eLife.106545.3.sa2
Reviewer #3 (Public review): https://doi.org/10.7554/eLife.106545.3.sa3
Author response https://doi.org/10.7554/eLife.106545.3.sa4

# Additional files

## Supplementary files

Supplementary file 1. Summary of demographic and potential predictors between groups for all survey 1 contributors who met the inclusion criteria. The demographic information included is for the full 105 participants who met the inclusion criteria. While all 105 met the inclusion criteria, seven participants left the question about age blank. As age was included as a predictor in the analysis of intent, the missing values were managed with listwise deletion, assuming missing at random, reducing the dataset size to 98 participants (N=35 baseline, N=48 interested, and N=15 intervention). To test for a statistically significant difference (association) between the treatment groups, a Pearson's Chi-square test was used for categorical variables, ordered logistic regression for nominal variables, and ANOVA test for continuous variables. Institute type was collected as a demographic, the resulted population sampled was predominately academic and this variable was therefore removed from downstream analysis due to the lack of predictive ability to assess institute type on the outcome of interest. The abbreviation name in bracket, within the demographic column, indicates the term used within the statistical model and associated output.

Supplementary file 2. Summary of demographic and potential predictors for all participants who met the inclusion criteria for Study 2. The demographic information is for the full 31 participants who met the inclusion criteria in study 2. While 31 unique individuals met the criteria, an individual may not have responded to both surveys (N=29 pre-survey and N=28 in the post-survey). For

instance, two participants did not meet the inclusion criteria for the pre-survey but met the criteria for the post-intervention survey. For the intention analysis, some missing data was observed in the demographic data. To reduce survey burden, the post-survey only included one demographic question (the participant's age) to support alignment of data, just in case duplicate initials were used as an identifier. Two responders did not include the age information in the study and were managed with listwise deletion, assuming missing at random. The abbreviation name in bracket, within the demographic column, indicates the term used within the statistical model and associated output. For McNemar's paired analysis, we conducted listwise deletion, assuming missing at random, reducing the dataset size to 26.

Supplementary file 3. Cumulative knowledge score for study 1. For each participant of study 1, the cumulative knowledge score is reported along with metadata of treatment group and reported gender. Participant identity is masked.

Supplementary file 4. Pre- and post-cumulative knowledge scores for study 2. For each participant in study 2, the pre- and post-cumulative knowledge score is reported. Participant identity is masked.

Supplementary file 5. Study 1 survey. The survey that was used in study 1 for all participants.

Supplementary file 6. Pre-workshop survey. The survey that was used in study 2 for all participants prior to the intervention.

Supplementary file 7. Postworkshop survey. The survey that was used in study 2 for all participants post the intervention.

Supplementary file 8. Workshop training material. The training material used in study 2.

Supplementary file 9. Cumulative knowledge score for study 1. For each participant of study 1, the cumulative knowledge score is reported alone with meta data of treatment group and reported gender. Participant identity is masked.

Supplementary file 10. Pre- and post-cumulative knowledge scores for study 2. For each participant of study 2, the pre and post cumulative knowledge score is reported. Participant identity is masked.

Supplementary file 11. Intention analysis for both study 1 and 2. The data and SAS code which can reproduce the analysis of intention for study 1 and study 2.

Supplementary file 12. Analysis of cumulative knowledge score for study 1. The Rmarkdown output from the analysis of the cumulative knowledge score for study 1.

MDAR checklist

## Data availability

All data generated during this research has been made available within the manuscript or via the supplementary information.

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

# Appendix 1

## Pearson's correlation coefficient analysis between continuous variables

Where Year_Work represents the number of years the participants have worked in animal research, Education represents the highest level of education obtained, Type_Work represents the type of research conducted by the participant, Training represents the level of statistical training received, Factorial_Fam represents how familiar the participants were with factorial experimental design, Factorial_Incor represents how often the participants incorporated males and females into their experiments while studying an intervention, attitude represents the average attitude score, Beh_control represents the average behavioural control score, Soc_norm represents the average social normal score and Ability_Influence represents how often the participants were involved or could influence the planning of experiments involving animals. Only variables that were used in the final statistical model were compared for correlations.

**Appendix 1—table 1.** Study 1.

|  | Ability_Influence | Factorial_Fam | Factorial_Incor | Attitude | Beh_Control | Soc_Norm |
|---|---|---|---|---|---|---|
| Ability_Influence | 1.0000 | 0.3499 | 0.2581 | –0.0138 | 0.2218 | 0.1514 |
| Factorial_Fam | 0.3499 | 1.0000 | 0.3317 | 0.2172 | 0.2371 | 0.2341 |
| Factorial_Incor | 0.2581 | 0.3317 | 1.0000 | 0.2539 | 0.4496 | 0.4384 |
| Attitude | –0.0138 | 0.2172 | 0.2539 | 1.0000 | 0.2081 | 0.1555 |
| Beh_Control | 0.2218 | 0.2371 | 0.4496 | 0.2081 | 1.0000 | 0.3691 |
| Soc_Norm | 0.1514 | 0.2341 | 0.4384 | 0.1555 | 0.3691 | 1.0000 |

**Appendix 1—table 2.** Study 2.

|  | Ability_Influence | Factorial_Fam | Factorial_Incor | Attitude | Beh_Control | Soc_Norm | Age |
|---|---|---|---|---|---|---|---|
| Ability_Influence | 1.0000 | 0.1361 | 0.1862 | 0.1305 | 0.3818 | 0.2926 | 1.0000 |
| Factorial_Fam | 0.1361 | 1.0000 | –0.2319 | 0.0714 | 0.3041 | 0.2671 | 0.1361 |
| Factorial_Incor | 0.1862 | –0.2319 | 1.0000 | –0.0689 | 0.1139 | 0.0407 | 0.1862 |
| Attitude | 0.1305 | 0.0714 | –0.0689 | 1.0000 | 0.2664 | 0.0710 | 0.1305 |
| Beh_Control | 0.3818 | 0.3041 | 0.1139 | 0.2664 | 1.0000 | 0.1624 | 0.3818 |
| Soc_Norm | 0.2926 | 0.2671 | 0.0407 | 0.0710 | 0.1624 | 1.0000 | 0.2926 |
| Age | 0.2047 | –0.0511 | –0.0616 | –0.3604 | 0.1078 | 0.1659 | 0.2047 |

## Appendix 2

### For the perceived barriers, the Pearson's chi-square test of association between treatment groups in study 1

**Appendix 2—table 1.** Test of association of the perceived barriers between treatment groups for study 1.

This survey question provided several pre-defined options and ability to enter a free-text option. Participants were asked to choose all that applied. Exploration of the free-text has grouped the barriers into three additional categories: convention, logistic, and none or no barriers. To test for a statistically significant difference (association) between the treatment groups, a Pearson's Chi-square test was applied for all options where the total N>10.

| Barrier offered | Total (count) possible N=105 | Baseline (count) possible N=39 | Interested (count) possible N=51 | Intervention (count) possible N=15 | Test of association test statistic & p-value |
|---|---|---|---|---|---|
| Cost | 44 | 14 | 24 | 6 | $X^2 = 1.157; p = 0.561$ |
| Male animals are more likely to fight and may lead to premature euthanasia | 34 | 8 | 21 | 5 | $X^2 = 4.317; p = 0.115$ |
| Female animals are more variable | 29 | 6 | 18 | 5 | $X^2 = 4.668; p = 0.097$ |
| Complexity of experimental design | 28 | 7 | 18 | 3 | $X^2 = 3.789; p = 0.149$ |
| Model behavior may be different in the other sex | 27 | 10 | 14 | 3 | $X^2 = 0.337; p = 0.844$ |
| Sample size concerns | 24 | 7 | 13 | 4 | $X^2 = 0.857; p = 0.651$ |
| Not relevant to the research question | 19 | 9 | 9 | 1 | $X^2 = 1.98; p = 0.371$ |
| Availability of sample/ test material | 12 | 2 | 7 | 3 | $X^2 = 2.884; p = 0.236$ |
| Welfare issues | 5 | 2 | 2 | 1 | NA |
| Data analysis concerns | 0 | 0 | 0 | 0 | NA |
| Other - Convention | 7 | 2 | 3 | 2 | NA |
| Other – Logistics | 1 | 1 | 0 | 0 | NA |
| Other - None | 3 | 1 | 0 | 2 | NA |

# Appendix 3

## For the perceived benefits, the Pearson's chi-square test of association between treatment groups in study 1

**Appendix 3—table 1.** Test of association of the perceived benefits between treatment groups for study 1.

This question provided several pre-defined options and the ability to enter a free-texted option. Participants were asked to choose all that applied. No free-text advantages were provided by survey takers for this question. To test for a statistically significant difference (association) between the treatment groups, a Pearson's chi-square test of association was applied for all options where the total N>10.

| Benefits offered | Total possible N=105 | Baseline (count) possible N=39 | Interested (count) possible N=51 | Intervention (count) possible N=15 | Test of association test statistic & p-value |
|---|---|---|---|---|---|
| Understanding sex differences | 91 | 33 | 44 | 14 | $X^2$=0.726; *p*=0.695 |
| Translatability | 79 | 29 | 35 | 15 | $X^2$=**6.149**; *p*=**0.046** |
| Efficient use of all animals from breeding | 42 | 12 | 18 | 12 | $X^2$=**11.855**; *p*=**0.0026** |
| Reproducibility | 42 | 17 | 16 | 9 | $X^2$=4.291; *p*=0.117 |
| 3Rs – Reduction | 33 | 11 | 14 | 8 | $X^2$=3.902; *p*=0.142 |
| Animal welfare | 22 | 10 | 7 | 5 | $X^2$=3.48; *p*=0.176 |
| Other | 0 | 0 | 0 | 0 | NA |

# Appendix 4

## For the perceived barriers, the McNemar's test of association between pre- and post-intervention in study 2

**Appendix 4—table 1.** Test of association of the perceived barriers between pre- and post-intervention for study 2.

This question provided several pre-defined options and the ability to enter a free-texted option. Participants were asked to choose all that applied. No free-text advantages were provided by survey takers for this question. To test for a statistically significant difference (association) between the treatment groups, a McNemar's test was applied for all options where the total N>10.

| Barrier offered | Pre-Intervention (count) Possible N=29 | Post-Intervention (count) Possible N=28 | McNemar's test Test statistic & p value |
|---|---|---|---|
| Cost | 15 | 8 | $X^2=3.57$; $P=0.060$ |
| Complexity of experimental design | 14 | 5 | $X^2=5.33$; $P=0.021$ |
| Sample size concerns | 13 | 4 | $X^2=4.45$; $P=0.035$ |
| Model behaviour may be different in the other sex | 12 | 2 | $X^2=5.44$; $P=0.020$ |
| Availability of sample/test material | 10 | 9 | $X^2=0.0$; $P=1.0$ |
| Female animals are more variable | 9 | 2 | $X^2=5.0$; $P=0.025$ |
| Experiment would take longer | 4 | 5 | NA |
| Male animals are more likely to fight and may lead to premature euthanasia | 3 | 2 | NA |
| Convention | 2 | 4 | NA |
| Data analysis concerns | 2 | 4 | NA |
| Logistics | 3 | 4 | NA |
| Welfare issues | 2 | 1 | NA |
| Not relevant to the research question | 0 | 4 | NA |
| Free-text: Sponsor request | 0 | 1 | NA |
| Free-text: Model induction issue | 0 | 1 | NA |

# Appendix 5

## For the perceived benefits, the McNemar's test of association between pre- and post-intervention in study 2

**Appendix 5—table 1.** Test of association of the perceived benefits between pre- and post-intervention for study 2.

This survey question provided several pre-defined options and ability to enter a free-texted option. Participants were asked to choose all that applied. Participants were given the option of a free-text response, but none were submitted. To test for a statistically significant difference (association) between the pre- and post-measures, a McNemar's test was applied for all options where the total N>10. This test accounts for the repeat nature of the data which required the list-wise deletion of those individuals with missing data.

| Advantages offered | Pre-intervention (count) possible N=29 | Post-intervention (count) possible N=28 | McNemar's test- statistic & p-value |
|---|---|---|---|
| Understanding sex differences | 26 | 25 | $X^2=0.0$; $p=1.0$ |
| Translatability | 21 | 22 | $X^2=1.0$; $p=0.32$ |
| Efficient use of all animals from breeding | 13 | 17 | $X^2=1.6$; $p=0.21$ |
| 3Rs - Reduction | 8 | 13 | $X^2=2.78$; $p=0.096$ |
| Reproducibility | 6 | 13 | $X^2=6.0$; $p=0.014$ |
| Animal welfare | 3 | 5 | NA |

## Appendix 6

### For the knowledge questions, a Pearson's chi-squared test to assess association between treatment groups of the proportion of participants who answered the question correctly in study 1

Appendix 6—table 1. Test of association for study 1 for each knowledge question.

| Question (Answer) | Total (Baseline & Interest) N=90 | Baseline (% correct) N=39 | Interest (% correct) N=51 | Intervention (% correct) N=15 | Test of association [Baseline v Interest] Test statistic & p-value | Test of association [across the 3 treatment groups] test statistic & p-value |
|---|---|---|---|---|---|---|
| Do you think inclusion of both sexes requires doubling a study's sample size? (No) | 20 (30.3%) | 13 (33%) | 7 (14%) | 10 (67%) | $X^2$=5.973; p=0.014 | $X^2$=16.85; p=0.00021 |
| Do you think sex influences data variability, therefore, when you include both sexes, more animals are needed? (No) | 9 (10%) | 5 (13%) | 4 (8%) | 6 (60%) | $X^2$ = 0.608; p=0.4355 | $X^2$=7.994; p=0.01837 |
| When analyzing in vivo data collected from both sexes, do you think sex should be included in the statistical model? (Yes) | 68 (75.6%) | 34 (87%) | 34 (67%) | 12 (80%) | $X^2$=5.035; p=0.0248 | $X^2$ = 5.266; p=0.072 |
| When analyzing in vivo data, do you think data from the two sexes should be pooled (combined) for an intervention into a single group for the analysis? (No) | 17 (18.9%) | 6 (15%) | 11 (21%) | 6 (40%) | $X^2$ = 0.552; p=0.457 | $X^2$ = 3.844; p=0.1463 |
| When analyzing in vivo data collected from both sexes, do you think the analysis should be run independently for each sex through separate statistical tests? (No) | 18 (20%) | 7 (18%) | 11 (21%) | 8 (53%) | $X^2$ = 0.181; p=0.671 | $X^2$=7.823; p=0.020 |

# Appendix 7

## For the knowledge questions, a McNemar's test was used to assess association between pre- and post-answers for each knowledge question in study 2

**Appendix 7—table 1.** Test of association for study 2 for each knowledge question.

| Question (Answer) | Pre (% correct) N=29 | Post (% correct) N=28 | McNemar's test of association test statistic & $p$-value |
|---|---|---|---|
| Do you think inclusion of both sexes requires doubling a study's sample size? (No) | 6 (21%) | 17 (61%) | $X^2$=7.14; $p$=0.007 |
| Do you think sex influences data variability, therefore, when you include both sexes, more animals are needed? (No) | 2 (7%) | 13 (46%) | $X^2$=9.31; $p$=0.0023 |
| When analyzing in vivo data collected from both sexes, do you think sex should be included in the statistical model? (Yes) | 21 (72%) | 21 (75%) | $X^2$ = 0.143; $p$=0.71 |
| When analyzing in vivo data collected from both sexes, do you think the analysis should be disaggregated (for example, run a t-test comparing control and intervention on the male data and then running a separate test on the data from females)? (No) | 8 (28%) | 22 (79%) | $X^2$=8.0; $p$=0.0047 |
| When analyzing in vivo data, do you think data from the two sexes should be pooled (combined) for the analysis (for example run a single t-test comparing control and intervention group)? (No) | 5 (17%) | 18 (64%) | $X^2$=10.29; $p$=0.0013 |

# Appendix 8

## Free-text response to barrier and benefit questions for studies 1 and 2

Participants had the opportunity to provide free-text responses to questions on the barriers and benefits of sex inclusive research. Only a few responses were received for the barrier question and are detailed below. No open-ended answers were provided for the benefits to sex inclusion.

**Appendix 8—table 1.** Study 1 free-text response.

| Statement | Classified | Group |
|---|---|---|
| Biomed staff generally keep the male rats for experimental use. Generally, I think because it is the way it has been done. | Other - Convention | Baseline |
| Convention | Other -Convention | Baseline |
| This is how my PI is doing it for many years | Other -Convention | Baseline |
| My supervisors | Other – Convention | Baseline |
| All historical data generated in make animals in a very narrow field | Other -Convention | Baseline |
| None | Other – None | Baseline |
| Logistics (duration of experiments) | Other - Logistics | Baseline |
| Larger male is preferred for implant of device | Other - Logistics | Baseline |
| After the workshop, I no longer have concerns about including both sexes in future experiments. | Other - None | Intervention |
| Expectation of boss! | Other - Convention | Intervention |
| Regulatory standard designs that are traditionally used | Other - Convention | Intervention |
| Cost of female animals | Other - Logistic | Intervention |
| None | Other – None | Intervention |

**Appendix 8—table 2.** Study 2 free-text response.

| Statement | Classified | Group |
|---|---|---|
| Availability of transgenics that are bred | Other - Logistic | Pre |
| Experimental model phenotype | Other - Logistic | Pre |
| We have to stagger groups in order to test all animals | Other - Logistic | Post |
| Disease exists in female (humans), however, experimental model in female mice protected | Other - Logistic | Post |
| Need a better understanding of stats - already look at genotype, time, and treatment intersections, not sure how to add sex as another variable | Other - Logistic | Post |
| Sponsor request | Other - Convention | Post |

## Appendix 9

## Glossary

Technical terms, used within this manuscript, and their associated definitions are detailed in the table below. For those terms associated to the theory of planned behaviour, the definitions provided were originally published by Icek Ajzen (*Ajzen, 1991*). The name in brackets represents the term used within the statistical model and associated output.

| Word | Definition |
|---|---|
| Theory of planned behaviour | An empirically supported framework that predicts an individual's intention to perform a particular behaviour by assessing that participant's attitude, subjective norms, and perceived behavioural control over the behaviour. |
| Intention (Avg_Intent) | The motivational factors that influence a given behaviour. The stronger the intention, the more likely the behaviour will be performed. |
| Attitude toward the behaviour (Attitude) | The degree to which a person believes a particular behaviour is good or bad. |
| Subjective norm (Soc_Norm) | This refers to the perceived social pressure experienced by an individual to engage, or not engage, in the behaviour in question. |
| Perceived behavioural control (Beh _Control) | This includes both actual and perceived control over the behaviour by the participant. Actual control covers resources and opportunities that are available to the participant. Perceived control can encompass an individual's perception about the ease or difficulty in performing the behaviour, which may encompass self-efficacy beliefs or confidence in conducting the behaviour. |

