## [Editor Report · eLife Assessment]

The authors quantified intentions and knowledge gaps in scientists' use of sex as a biological variable in their work, and used a workshop intervention to show that while willingness was high, pressure points centered on statistical knowledge and perceived additional monetary costs to research. These **important** findings demonstrate the difficulty in changing understanding: while interventions can improve knowledge and decrease perceived barriers, the impact was small. The evidence for the findings is **solid**.

---

## [Referee Report · Reviewer #1 (Public review)]

Summary:

The authors use the theory of planned behavior to understand whether or not intentions to use sex as a biological variable (SABV), as well as attitude (value), subjective norm (social pressure), and behavioral control (ability to conduct behavior), across scientists at a pharmacological conference. They also used an intervention (workshop) to determine the value of this workshop in changing perceptions and misconceptions. Attempts to understand the knowledge gaps were made.

Strengths:

The use of SABV is limited in terms of researchers using sex in the analysis as a variable of interest in the models (and not a variable to control). To understand how we can improve on the number of researchers examining the data with sex in the analyses, it is vital we understand the pressure points that researchers consider in their work. The authors identify likely culprits in their analyses. The authors also test an intervention (workshop) to address the main bias or impediments for researchers' use of sex in their analyses.

---

## [Referee Report · Reviewer #2 (Public review)]

Summary:

The investigators tested a workshop intervention to improve knowledge and decrease misconceptions about sex inclusive research.

Strengths:

The investigators included control groups and replicated the study in a second population of scientists. The results appear to be well substantiated. Figures are easy to understand.

Weaknesses: None noted

Comments on revised version:

The authors have responded appropriately to all of my concerns.

---

## [Referee Report · Reviewer #3 (Public review)]

Summary:

This manuscript aims to determine cultural biases and misconceptions in inclusive sex research and evaluate the efficacy of interventions to improve knowledge and shift perceptions to decrease perceived barriers for including both sexes in basic research.

Overall, this study demonstrates that despite the intention to include both sexes and a general belief in the importance of doing so, relatively few people routinely include both sexes. Further, the perceptions of barriers to doing so are high, including misconceptions surrounding sample size, disaggregation, and variability of females. There was also a substantial number of individuals without the statistical knowledge to appropriately analyze data in studies inclusive of sex. Interventions increased knowledge and decreased perception of barriers.

Strengths:

(1) This manuscript provides evidence for the efficacy of interventions for changing attitudes and perceptions of research.

(2) This manuscript also provides a training manual for expanding this intervention to broader groups of researchers.

---

## [Author Response]

The following is the authors’ response to the original reviews.

**Reviewer #1 (Public review):**
Summary:The authors use the theory of planned behavior to understand whether or not intentions to use sex as a biological variable (SABV), as well as attitude (value), subjective norm (social pressure), and behavioral control (ability to conduct behavior), across scientists at a pharmacological conference. They also used an intervention (workshop) to determine the value of this workshop in changing perceptions and misconceptions. Attempts to understand the knowledge gaps were made.Strengths:The use of SABV is limited in terms of researchers using sex in the analysis as a variable of interest in the models (and not a variable to control). To understand how we can improve on the number of researchers examining the data with sex in the analyses, it is vital we understand the pressure points that researchers consider in their work. The authors identify likely culprits in their analyses. The authors also test an intervention (workshop) to address the main bias or impediments for researchers' use of sex in their analyses.Weaknesses:There are a number of assumptions the authors make that could be revisited:(1) that all studies should contain across sex analyses or investigations. It is important to acknowledge that part of the impetus for SABV is to gain more scientific knowledge on females. This will require within sex analyses and dedicated research to uncover how unique characteristics for females can influence physiology and health outcomes. This will only be achieved with the use of female-only studies. The overemphasis on investigations of sex influences limits the work done for women's health, for example, as within-sex analyses are equally important.

The Sex and Gender Equity in Research (SAGER) guidelines (1) provide guidance that “Where the subjects of research comprise organisms capable of differentiation by sex, the research should be designed and conducted in a way that can reveal sex-related differences in the results, even if these were not initially expected.”. This is a default position of inclusion where the sex can be determined and analysis assessing for sex related variability in response. This position underpins many of the funding bodies new policies on inclusion.

However, we need to place this in the context of the driver of inclusion. The most common reason for including male and female samples is for those studies that are exploring the effect of a treatment and then the goal of inclusion is to assess the generalisability of the treatment effect (exploratory sex inclusion)(2). The second scenario is where sex is included because sex is one of the variables of interest and this situation will arise because there is a hypothesized sex difference of interest (confirmatory sex inclusion).

We would argue that the SABV concept was introduced to address the systematic bias of only studying one sex when assessing treatment effect to improve the generalisability of the research. Therefore, it isn’t directly to gain more scientific knowledge on females. However, this strategy will highlight when the effect is very different between male and female subjects which will potentially generate sex specific hypotheses.

Where research has a hypothesis that is specific to a sex (e.g. it is related to oestrogen levels) it would be appropriate to study only the sex of interest, in this case females. The recently published Sex Inclusive Research Framework gives some guidance here and allows an exemption for such a scenario classifying such proposals “Single sex study justified” (3).

We have added an additional paragraph to the introduction to clarify the objectives behind inclusion and how this assists the research process.

(2) It should be acknowledged that although the variability within each sex is not different on a number of characteristics (as indicated by meta-analyses in rats and mice), this was not done on all variables, and behavioral variables were not included. In addition, across-sex variability may very well be different, which, in turn, would result in statistical sex significance. In addition, on some measures, there are sex differences in variability, as human males have more variability in grey matter volume than females. PMID: 33044802.

The manuscript was highlighting the common argument used to exclude the use of females, which is that females are inherently more variable as an absolute truth. We agree there might be situations, where the variance is higher in one sex or another depending on the biology. We have extended the discussion here to reflect this, and we also linked to the Sex Inclusive Research Framework (3) which highlights that in these situations researchers can utlise this argument provided it is supported with data for the biology of interest.

(3) The authors need to acknowledge that it can be important that the sample size is increased when examining more than one sex. If the sample size is too low for biological research, it will not be possible to determine whether or not a difference exists. Using statistical modelling, researchers have found that depending on the effect size, the sample size does need to increase. It is important to bare this in mind as exploratory analyses with small sample size will be extremely limiting and may also discourage further study in this area or indeed as seen the literature - an exploratory first study with the use of males and females with limited sample size, only to show there is no "significance" and to justify this as an reason to only use males for the further studies in the work.

The reviewer raises a common problem: where researchers have frequently argued that if they find no sex differences in a pilot then they can proceed to study only one sex. The SAGER guidelines (1), and now funder guidelines (4, 5), challenge that position. Instead, the expectation is for inclusion as the default in all experiments (exploratory inclusion strategy) to allow generalisable results to be obtained. When the results are very different between the male and female samples, then this can be determined. This perspective shift (2) requires a change in mindset and understanding that the driver behind inclusion is of generalisability not exploration of sex differences. This has been added to the introduction as an additional paragraph exploring the drivers behind inclusion.

We agree with the reviewer that if the researcher is interested in sex differences in an effect (confirmatory inclusion strategy, aka sex as a primary variable) then the N will need to be higher. However, in this situation, one, of course, must have male and female samples in the same experiment to allow the simultaneous exploration to assess the dependency on sex.

**Reviewer #2 (Public review):**
Summary:The investigators tested a workshop intervention to improve knowledge and decrease misconceptions about sex inclusive research. There were important findings that demonstrate the difficulty in changing opinions and knowledge about the importance of studying both males and females. While interventions can improve knowledge and decrease perceived barriers, the impact was small.Strengths:The investigators included control groups and replicated the study in a second population of scientists. The results appear to be well substantiated. These are valuable findings that have practical implications for fields where sex is included as a biological variable to improve rigor and reproducibility.

Thank you for assessment and highlighting these strengths. We appreciate your recognition of the value and practical implications of this work.

Weaknesses:I found the figures difficult to understand and would have appreciated more explanation of what is depicted, as well as greater space between the bars representing different categories.

We have improved the figures and figure legends to improve clarity.

**Reviewer #3 (Public review):**
Summary:This manuscript aims to determine cultural biases and misconceptions in inclusive sex research and evaluate the efficacy of interventions to improve knowledge and shift perceptions to decrease perceived barriers for including both sexes in basic research.Overall, this study demonstrates that despite the intention to include both sexes and a general belief in the importance of doing so, relatively few people routinely include both sexes. Further, the perceptions of barriers to doing so are high, including misconceptions surrounding sample size, disaggregation, and variability of females. There was also a substantial number of individuals without the statistical knowledge to appropriately analyze data in studies inclusive of sex. Interventions increased knowledge and decreased perception of barriers.Strengths:(1) This manuscript provides evidence for the efficacy of interventions for changing attitudes and perceptions of research.(2) This manuscript also provides a training manual for expanding this intervention to broader groups of researchers.

Thank you for highlighting these strengths. We appreciate your recognition that the intervention was effect in changing attitudes and perception. We deliberately chose to share the material to provide the resources to allow a wider engagement.

Weaknesses:The major weakness here is that the post-workshop assessment is a single time point, soon after the intervention. As this paper shows, intention for these individuals is already high, so does decreasing perception of barriers and increasing knowledge change behavior, and increase the number of studies that include both sexes? Similarly, does the intervention start to shift cultural factors? Do these contribute to a change in behavior?

Measuring change in behaviour following an intervention is challenging and hence we had implemented an intention score as a proxy for behaviour. We appreciate the benefit of a long-term analysis, but it was beyond the scope of this study and would need a larger dataset size to allow for attrition. We agree that the strategy implemented has weaknesses. We have extended the limitation section in the discussion to include these.

**Reviewer #1 (Recommendations for the authors):**
I would ask them to think about alternative explanations and ask for free-form responses, and to revise with the caveats written above - sample size does need to be increased depending on effect size, and that within sex studies are also important. Not all studies should focus on sex influences.

The inclusion of the additional paragraph in the introduction to clarify the objective of inclusion and the resulting impact on experimental design should address these recommendations.

We have also added the free-form responses as an additional supplementary file.

**Reviewer #2 (Recommendations for the authors):**
This is an important set of studies. My only recommendation to improve the data presentation so that it is clear what is depicted and how the analyses were conducted. I know it is in the methods, but reminding the reader would be helpful.

We have revisited the figures and included more information in the legends to explain the analysis and improve clarity.

**Reviewer #3 (Recommendations for the authors):**
There are parts in the introduction which read as contradictory and as such are confusing - for example, in the 3rd paragraph it states that little progress on sex inclusive research has been made, and in the following sentences it states that the proportion of published studies across sex has improved. The references in these two statements are from the same time range, so has this improved? Or not?

The introduction does include a summation statement on the position: “Whilst a positive step forward, this proportion still represents a minority of studies, and notably this inclusion was not associated with an increase in the proportion of studies that included data analysed by sex.” We have reworded the text to ensure it is internally consistent with this summary statement and this should increase clarity.

In discussing the results, it is sometimes confusing what the percentages mean. For example, "the researchers reported only conducting sex inclusive research in <= 55% of their studies over the past 5 years (55% in study 1 general population and 35% study 2 pre-assessment)." Does that mean 55% of people are conducting sex inclusive research, or does this mean only half of their studies? These two options have very different implications.

We agree that the sentence is confusing and it has been reworded.

Addressing long-term assessments in attitude and action (ie, performing sex inclusive research) is a crucial addition, with data if possible, but at least substantive discussion.

We have add this to the limitation section in the discussion

One minor but confusing point is the analogy comparing sex inclusive studies with attending the gym. The point is well taken - knowledge is not enough for behavior change. However, the argument here is that to increase sex inclusive research requires cultural change. To go to the gym, requires motivation.This seems like an oranges-to-lemons comparison (same family, different outcome when you bite into it).

At the core, both scenarios involve the challenge of changing established habits and cultural norms in action based on knowledge (the right thing to do). The exercise scenario is a primary example provided by the original authors to describe how aspects of the theory of planned behaviour (perceived behavioural control, attitude, and social norms) may influence behavioural change. Understanding which of these aspects may drive or influence change is why we used this framework to understand our study population. We disagree that is an oranges-to-lemons comparison.

References

(1) Heidari S, Babor TF, De Castro P, Tort S, Curno M. Sex and Gender Equity in Research: rationale for the SAGER guidelines and recommended use. Res Integr Peer Rev. 2016;1:2.

(2) Karp NA. Navigating the paradigm shift of sex inclusive preclinical research and lessons learnt. Commun Biol. 2025;8(1):681.

(3) Karp NA, Berdoy M, Gray K, Hunt L, Jennings M, Kerton A, et al. The Sex Inclusive Research Framework to address sex bias in preclinical research proposals. Nat Commun. 2025;16(1):3763.

(4) Medical Research Council. Sex in experimental design - Guidance on new requirements https://www.ukri.org/councils/mrc/guidance-for-applicants/policies-and-guidance-forresearchers/sex-in-experimental-design/: UK Research and Innovation; 2022

(5) Clayton JA, Collins FS. Policy: NIH to balance sex in cell and animal studies. Nature. 2014;509(7500):282-3.